# Prevalence of Alzheimer's disease pathology in the community

Dag Aarsland[1,2 ✉], Anita Lenora Sunde[1,3 ✉], Diego A. Tovar-Rios[1,4,5,6], Antoine Leuzy[1], Tormod Fladby[7], Henrik Zetterberg[8,9,10,11,12,13], Kaj Blennow[8,13,14,15], Kübra Tan[8], Giovanni De Santis[8], Yara Yakoub[16], Burak Arslan[8], Hanna Huber[8,17], Ilaria Pola[8], Lana Grötschel[8], Guglielmo Di Molfetta[8], Håvard K. Skjellegrind[18,19], Geir Selbaek[20,21,22] & Nicholas J. Ashton[8,23,24]

The prevalence of Alzheimer's disease neuropathological changes (ADNCs), the leading cause of cognitive impairment, remains uncertain. Recent blood-based biomarkers enable scalable assessment of ADNCs[1]. Here we measured phosphorylated tau at threonine 217 in 11,486 plasma samples from a Norwegian population-based cohort of individuals over 57 years of age as a surrogate marker for ADNCs. The estimated prevalence of ADNCs increased with age, from less than 8% in people 58–69.9 years of age to 65.2% in those over 90 years of age. Among participants aged 70 years or older, 10% had preclinical Alzheimer's disease, 10.4% had prodromal Alzheimer's disease and 9.8% had Alzheimer's disease dementia. Furthermore, among those 70 years of age or older, ADNCs were present in 60% of people with dementia, in 32.6% of those with mild cognitive impairment and in 23.5% of the cognitively unimpaired group. Our findings suggest a higher prevalence of Alzheimer's disease dementia in older individuals and a lower prevalence of preclinical Alzheimer's disease in younger groups than previously estimated[2].

Dementia is a growing global challenge, with Alzheimer's disease (AD) being the most common cause, characterized by ADNCs, encompassing brain deposits of amyloid-β (Aβ) plaque and neurofibrillary tau tangles. The prevalence of dementia and mild cognitive impairment (MCI) is well established[2,3], but the prevalence of ADNCs in general populations remains uncertain. With the advent of drugs capable of reducing Aβ plaque pathology and slowing cognitive decline[4,5], accurate knowledge of ADNC prevalence is essential for anticipating the number of individuals eligible for treatment and estimating future health-care demands and associated costs. A recent review has reported an overall prevalence of 22% ADNCs in all people 50 years of age and older globally[2]. However, studies examining the prevalence of ADNCs are typically enriched, including relatively small clinic-based samples, which tend to differ regarding important clinical and demographic features compared with general populations. Such studies may thus report inflated or deflated rates of AD pathology.

Until recently, ADNCs could only be verified in vivo using cerebrospinal fluid analysis or molecular positron emission tomography (PET), substantially hindering its evaluation in large population-based studies. Minimally invasive blood-based markers, particularly plasma phosphorylated tau at threonine 217 (pTau217), that have high accuracy for ADNCs have recently become available but have not yet been used in large community-based studies[6]. In this study, we capitalized on the large Norwegian population-based Trøndelag Health (HUNT) study[7,8], with 11,486 blood samples of participants 58 years of age and older, to explore the following research questions: (1) what the prevalence of ADNCs in the population 58 years of age and older across age and sex groups is; (2) what the association between ADNCs and demographics, cognition, educational level, apolipoprotein E (*APOE*) ε2, ε3 or ε4 status and comorbidities is; and (3) what proportion of those 70 years of age or older is eligible for disease-modifying therapies (DMTs) according to current recommendations[9,10].

The HUNT study has been ongoing for four decades, with a new wave taking place in the same population every 10 years, thus four waves exist so far. In this nested cross-sectional study, we included 2,537 individuals from HUNT3 (age range of 58–69.9 years, 51.2% women) and 8,949 from

[1]Centre for Age-Related Medicine (SESAM), Stavanger University Hospital, Stavanger, Norway. [2]Centre for Healthy Brain Ageing, Department of Psychological Medicine, Institute of Psychiatry, Psychology, and Neuroscience, King's College London, London, UK. [3]Department of Clinical Medicine, University of Bergen, Bergen, Norway. [4]L-BioStat, KU Leuven, Leuven, Belgium. [5]Grupo de Investigación en Estadística Aplicada- INFERIR, Universidad del Valle, Santiago de Cali, Colombia. [6]Prevención y Control de la Enfermedad Crónica- PRECEC, Universidad del Valle, Santiago de Cali, Colombia. [7]Department of Neurology, Akershus University Hospital, Lorenskog, Norway. [8]Department of Psychiatry and Neurochemistry, Institute of Neuroscience and Physiology, The Sahlgrenska Academy, University of Gothenburg, Gothenburg, Sweden. [9]Department of Neurodegenerative Disease, UCL Institute of Neurology, London, UK. [10]UK Dementia Research Institute at UCL, London, UK. [11]Hong Kong Center for Neurodegenerative Diseases, The Hong Kong University of Science and Technology, Hong Kong, China. [12]Wisconsin Alzheimer's Disease Research Center, University of Wisconsin School of Medicine and Public Health, University of Wisconsin-Madison, Madison, WI, USA. [13]Clinical Neurochemistry Laboratory, Sahlgrenska University Hospital, Gothenburg, Sweden. [14]Paris Brain Institute, ICM, Pitié-Salpêtrière Hospital, Sorbonne University, Paris, France. [15]Neurodegenerative Disorder Research Center, Division of Life Sciences and Medicine, Department of Neurology, Institute on Aging and Brain Disorders, University of Science and Technology of China and First Affiliated Hospital of USTC, Hefei, P. R. China. [16]Douglas Mental Health University Institute, Centre for Studies on the Prevention of Alzheimer's Disease, Montreal, Quebec, Canada. [17]German Center of Neurodegenerative Diseases (DZNE), Bonn, Germany. [18]HUNT Research Centre, Department of Public Health and Nursing, NTNU, Norwegian University of Science and Technology, Levanger, Norway. [19]Levanger Hospital, Nord-Trøndelag Hospital Trust, Levanger, Norway. [20]Norwegian National Centre for Aging and Health, Vestfold Hospital Trust, Tønsberg, Norway. [21]Department of Geriatric Medicine, Oslo University Hospital, Nydalen, Oslo, Norway. [22]Institute of Clinical Medicine, University of Oslo, Oslo, Norway. [23]Banner Sun Health Research Institute, Sun City, AZ, USA. [24]Banner Alzheimer's Institute and University of Arizona, Phoenix, AZ, USA. ✉e-mail: daarsland@gmail.com; anitalenora@gmail.com

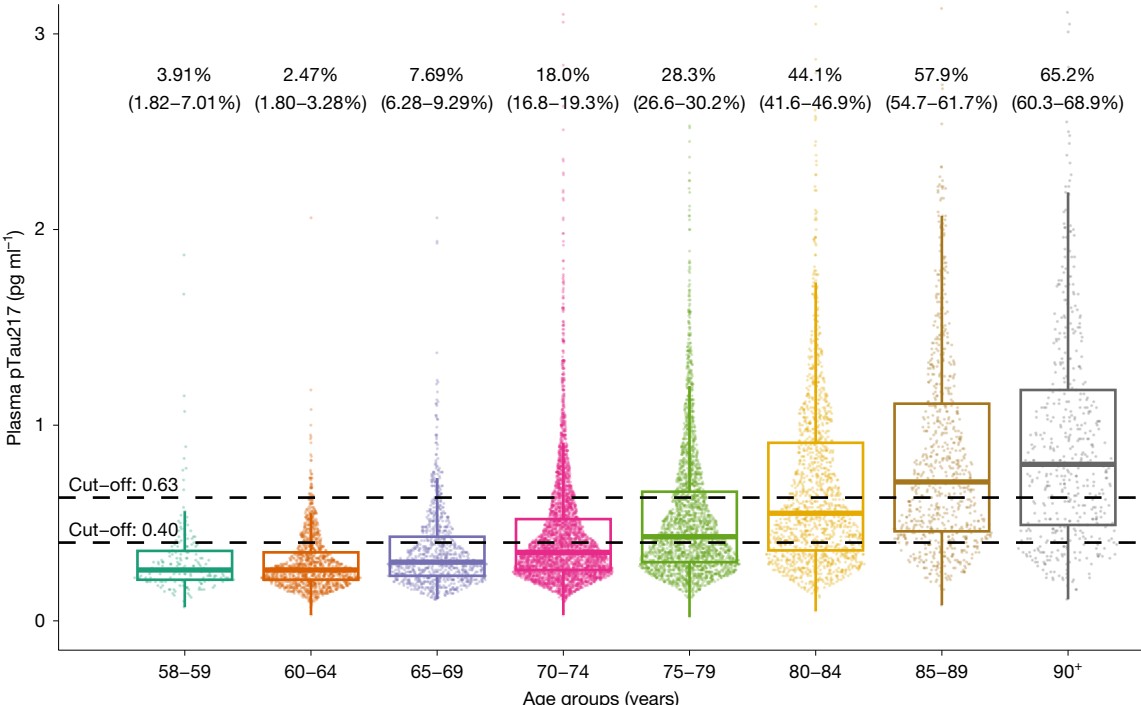

**Fig. 1 | Plasma pTau217 concentrations in different age groups.** Individual dots represent plasma pTau217 concentrations ($n$ = 2,537 participants from HUNT3 and $n$ = 8,949 participants from HUNT4 70+). Percentages (95% confidence interval) are estimates of how many in each age group have AD neuropathology, defined by plasma pTau217 concentration of 0.63 pg ml$^{-1}$ or more. The lower cut-off of 0.40 pg ml$^{-1}$ is also shown. The horizontal line in each box represents the median, and bottom and top edges delineate the second and third quartiles. The bottom whisker represents the first quartile, and the top whisker denotes the fourth quartile. Concentrations above 3 pg ml$^{-1}$ are not shown.

HUNT4 (age range of 70 years and older (hereafter the 70+ group), 53.6% women). Subsequent diagnostic history was considered when deciding who to approach for inclusion in HUNT3. Although a blood sample was provided in both surveys, the HUNT4 70+ cohort also underwent a standardized clinical assessment for a diagnosis of dementia and MCI[11] (Extended Data Fig. 1 and Supplementary Information, 'Assessment of cognition, physical performance, anxiety, depression, neuropsychiatric symptoms and activities of daily living'). The presence of ADNCs was established by measuring plasma pTau217 levels with a previously validated commercial kit (ALZpath p-Tau 217 Advantage PLUS, Quanterix)[1]. We used a two cut-off approach as recommended by the Global CEO Initiative on Alzheimer's Disease[12] to categorize individuals as ADNC negative (less than 0.40 pg ml$^{-1}$), intermediate or positive (0.63 pg ml$^{-1}$ or more), as previously described[1]. The agreement between elevated plasma pTau217 concentration and the presence of notable amounts of plaques and tangles at post-mortem examination has been previously found to be very high[13]. For terminological clarity, in the remainder of this article, the term 'ADNC' refers specifically to the presence of elevated plasma pTau217 concentration, used as a surrogate marker for ADNCs.

The demographic and clinical characteristics of the cohort are shown in Extended Data Table 1. The estimated proportion of people with and without ADNCs in different age groups is shown in Fig. 1 and Extended Data Table 2. There was a stepwise increase in the proportion of ADNCs across age groups; the proportion was 33.4% in the 70+ group. There was a significant association between ADNCs and cognitive diagnosis. Among individuals with dementia, 60% had ADNCs (that is, AD dementia), compared with 32.6% of those with MCI (that is, prodromal AD) and 23.5% in the cognitively unimpaired group (that is, preclinical AD). The proportion with ADNCs increased with age in each of the cognitive groups. ADNCs were ruled out based on plasma pTau217 concentrations being below the lower cut-off in 19.4% of the dementia group, 41% of the MCI group and 50.1% of the

cognitively unimpaired group (Fig. 2 and Extended Data Table 3). Depending on age, 13.5–27.6% of participants had plasma pTau217 concentrations in the intermediate range, with only minor differences between the cognitively unimpaired, MCI and dementia groups (Extended Data Tables 2 and 3). Weighted estimated proportions of ADNCs in the respective clinical cognitive subgroups are shown in Extended Data Table 4.

The estimated prevalence of preclinical AD, prodromal AD and AD dementia in the 70+ study population is shown in Fig. 3 and Extended Data Table 5. The estimated prevalence of AD dementia consistently increased with age, whereas the prevalence of preclinical AD increased from the 70–74 year age group to the 80–84 year age group before decreasing in the oldest old (those aged 85 years or older (the 85+ group)). The prevalence of prodromal AD remained stable after 80 years of age (Extended Data Fig. 2).

In the 80–89 year age group, men had a slightly higher estimated prevalence of ADNCs than women, mainly due to a higher prevalence of early-stage AD (preclinical and prodromal AD), although cognitive subgroup differences were not statistically significant. There was no sex difference in the estimated prevalence of AD dementia in any of the age groups. Details can be found in Extended Data Table 5 and Extended Data Fig. 2.

Estimated ADNC prevalence was higher among individuals with one (46.4%) or two (64.6%) *APOE* ε4 alleles than in those with none (27.1%; Extended Data Table 6). Estimated glomerular filtration rate (eGFR) was inversely associated with plasma pTau217 concentration in individuals with eGFR < 51 ml min$^{-1}$ per 1.73 m$^2$ (Extended Data Fig. 3). There was no significant association between ADNCs and self-reported cardiovascular and cerebrovascular disease, chronic obstructive pulmonary disease, diabetes, cancer, migraine, psoriasis, kidney disease, rheumatoid arthritis and gout, when adjusting for age, sex, *APOE* ε4 allele count, cognition, serum creatinine levels and education level (Extended Data Table 7).

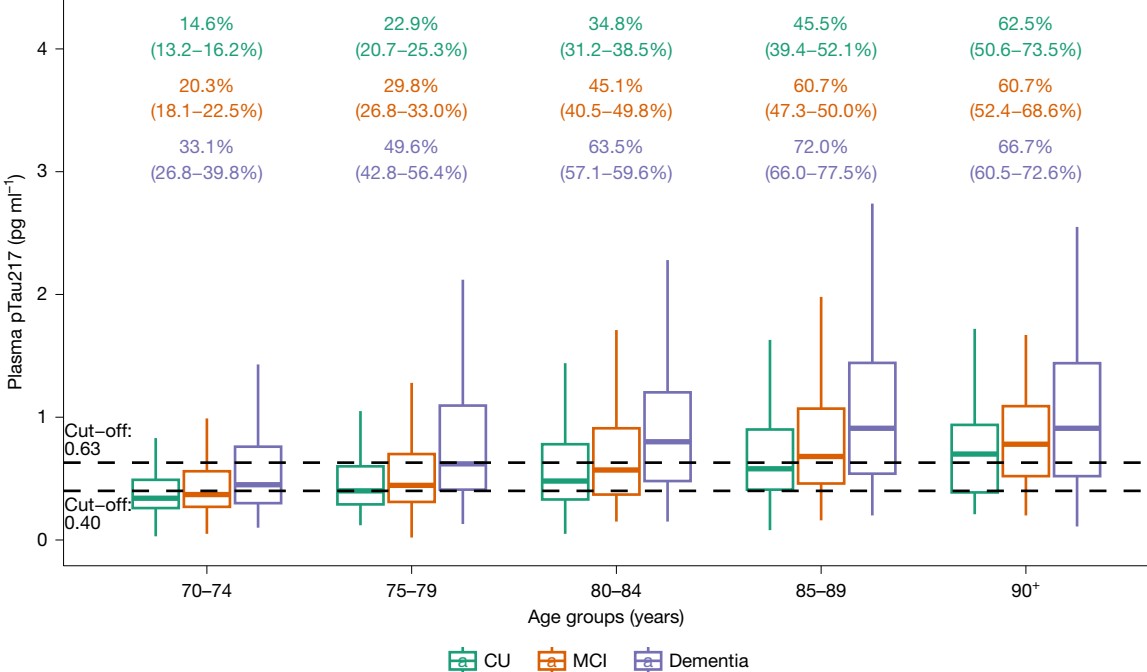

**Fig. 2 | Plasma pTau217 concentrations in people 70 years of age or older who are cognitively unimpaired, have MCI or have dementia.** Percentages (with 95% confidence intervals; colour-coded to match the box plots) are estimates of how many in each cognitive group have AD neuropathology, defined by plasma pTau217 concentration ≥ 0.63 pg ml⁻¹. The lower cut-off of 0.40 pg ml⁻¹ is also shown. $n$ = 8,949 participants from HUNT4 70+. The horizontal line in each box represents the median, and top and bottom edges delineate the second and third quartiles. The bottom whisker represents the first quartile, and the top whisker denotes the fourth quartile. CU, cognitively unimpaired.

Estimated ADNC prevalence was lowest among individuals with tertiary education, highest among those with primary education and intermediate in those with secondary education, with differences becoming more pronounced with age. For those with primary education, ADNC prevalence was higher in women than in men, whereas in those with secondary and tertiary education, men had a higher estimated ADNC prevalence than women (Extended Data Fig. 4).

On the basis of the current eligibility criteria for DMTs[4,5], out of a total of 8,949 participants in the 70+ group, 909 (10.2%) fulfilled the eligibility criteria for DMTs (Extended Data Fig. 5), whereas the weighted estimate for the entire 70+ population was 11.1%.

Positive and negative predictive values (PPVs and NPVs, respectively) were examined in an exploratory analysis based on previously reported sensitivity and specificity metrics for pTau217 for ADNCs. The PPV increased with age-specific prevalence, from 59.9% (70–74 years of age) to 92.6% (90 years of age or older (the 90+ group)), whereas the NPV decreased from 97.9% to 84.9%. Overall, the PPV and NPV were 77.4% and 95.4%, respectively. Under optimism correction, the PPV and NPV were attenuated to 71.9% and 92.9% overall, with ranges of 52.9–90.4% (for PPV) and 78.3–96.8% (for NPV) across age strata, consistent with higher PPV and lower NPV at higher prevalence (Extended Data Table 8).

In this large, Norwegian population-based cohort study, the proportion of individuals with ADNCs, as measured with plasma pTau217 concentration, increased with age, from under 8% in people 58–69 years of age to 65% in those over 90 years of age. Among the 70+ population, 10% had preclinical AD, 10.4% had prodromal AD and 9.8% had AD dementia. ADNCs were more prevalent in individuals with lower education and those with *APOE* ε4 alleles. ADNCs were inversely associated with eGFR < 51 ml min⁻¹ per 1.73 m². It has previously been shown that age, *APOE* ε4 status and renal dysfunction are associated with pTau217 concentrations, but that these factors only marginally alter the clinical performance of pTau217 as a marker for ADNCs[14,15]. The two plasma pTau217 cut-offs used to assess the prevalence of ADNCs

(lower cut-off with 95% sensitivity and upper cut-off with 95% specificity) were applied independent of age[1]. Age-dependent increases in plasma pTau, independent of ADNCs, have been discussed in the literature; however, current evidence does not support this association[6].

Although the prevalence of AD dementia among the youngest cognitively assessed age group in our study (70–74 years) was similar to that reported in a recent literature review (2% versus 1.7%)[2], we found a higher prevalence of AD dementia in the older age groups (for example, 85–89 years: 25.2% in our study, compared with 7.1% in the review). By contrast, the prevalence of preclinical AD in the youngest age group (70–74 years) was much higher in the review (22.4% for men and 22.2% for women) than in our study (7.6% for men and 7.9% for women). It is possible that, compared with the unselected community-based cohort in our study, the more selected cohorts included in the review over-recruited cognitively healthy people at high risk for AD, and under-recruited older people with dementia. Participants under 70 years of age were not cognitively assessed; thus, their clinical status remains undetermined.

Blood-based AD biomarkers are increasingly being utilized for considering people eligible for treatment with the new anti-AD drugs, which have been recently approved in several countries, including the USA and Europe. On the basis of the current eligibility recommendations for these drugs[9,10], we found that 11% of people 70 years of age or older in the study population would potentially be eligible. Such treatments carry risks that would have to be carefully weighed against any potential benefits in each individual[16].

On the basis of having plasma pTau217 concentrations below the lower cut-off, ADNCs were ruled out in 41% of the MCI group and 19.4% of the dementia group. Thus, in these groups, cognitive impairment is probably due to causes other than AD. The higher prevalence of ADNCs in people with one or two than with none *APOE* ε4 alleles is in line with previous findings[17,18]. The prevalence of ADNCs was slightly higher in men than in women in the 80–89 year age group. There was no sex

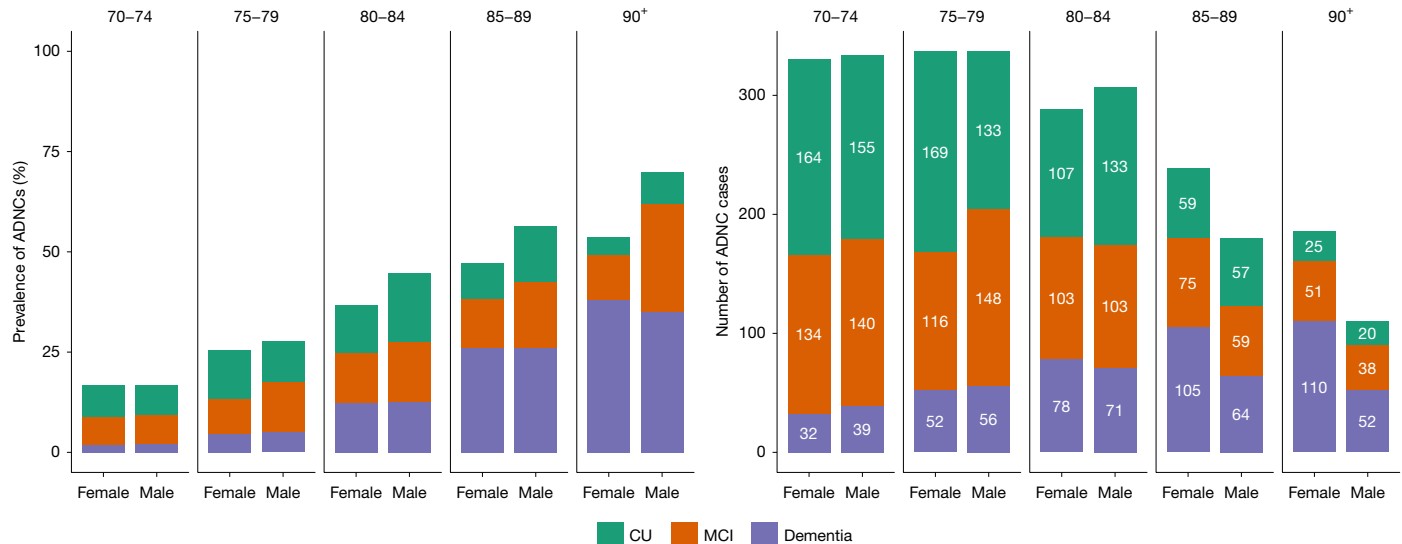

**Fig. 3 | Proportions and frequencies of ADNCs across the AD continuum in the 70+ population.** Left, the percentage of participants with ADNCs, defined as plasma pTau217 concentration ≥ 0.63 pg ml$^{-1}$. Stacked bars represent the estimated proportions of ADNCs. Right, absolute numbers of study participants with ADNCs. Colours represent different levels of cognitive effects. The values are stratified, on the $x$ axis, by sex. The numbers displayed at the top of the graphs are age groups in years.

difference in the prevalence of plasma pTau217-verified AD dementia in any age group. Several clinical studies have reported a female predominance of dementia[19], and previous HUNT data have reported a slightly higher prevalence of dementia in women than in men, but only in those over 85 years of age, whereas MCI was more common in men than in women[11]. The recent review also reported a higher proportion of preclinical AD in men, but a higher prevalence of AD dementia in women[2]. Thus, the most recent evidence does not seem to support previous reports of higher ADNC prevalence in women.

Having a lower level of education was clearly associated with higher ADNC prevalence, especially in the older age groups. This supports the theory of a protective effect of education, for example, by means of increasing cognitive reserve[20]. We did not assess potential confounding factors such as smoking, obesity, physical inactivity and excessive alcohol consumption[21], which may attenuate the association between education level and ADNCs.

There was no association between self-reported somatic morbidities and plasma pTau217 concentrations above the upper cut-off when adjusting for confounding factors. Previous research has shown that vascular disease can both cause dementia and increase the odds that AD pathology manifests as AD dementia[22,23]. We have not studied the possible synergistic effect of ADNCs and comorbidities on cognition.

This study has several strengths. It is the largest population-based study of ADNCs. The clinical diagnosis of MCI and dementia was based on a prospective standardized clinical assessment. Analysis of ADNCs was based on an accurate assay performed in a laboratory with extensive experience and expertise using state-of-the art technologies. Until recently, the absence of robust and scalable pTau217 assays meant that research cut-offs for blood-based markers were specific to each cohort and did not generalize effectively during external validation. However, the assay used in this study has shown strong performance in external validation across independent cohorts[1].

Plasma pTau217 reflects both phosphorylated, soluble tau in the context of Aβ pathology[24] and aggregated tau pathology. Elevated concentrations of soluble pTau217 can occur decades before the onset of aggregated tau[25], correlate strongly with the severity of AD pathology and with clinical progression, and are considered specific for AD[1]. Thus, plasma pTau217 has shown good discriminative accuracy for distinguishing between pathology-confirmed AD and other tauopathies such as frontotemporal lobar degeneration[26,27], traumatic encephalopathy syndrome[28], primary age-related tauopathy[29] and progressive supranuclear palsy[13].

Plasma pTau217 concentration has been shown to have the highest predictive values of the blood-based AD markers[30]. Although recent studies indicate that the ratio of plasma pTau217 to the 42-amino-acid-long form of Aβ (Aβ42) can reduce intermediate test results and may slightly increase diagnostic accuracy in well-controlled research cohorts[31–33], these findings do not directly translate to large population-based studies such as HUNT. Plasma Aβ42 concentration is highly sensitive to pre-analytical variables, which often deviate from Alzheimer's Association guidelines in population cohorts[34], leading to falsely increased pTau217:Aβ42 ratios[35].

This study used a two-cut-off method[1]. Depending on age, 13.5–27.6% of the study population had plasma pTau217 concentrations in the intermediate range, here defined as plasma pTau217 values between 0.40 and 0.63 pg ml$^{-1}$ and would thus require further examination to clarify their ADNC status. In an ideal scenario, individuals in the intermediate group would undergo cerebrospinal fluid analysis or PET imaging to obtain a more definitive diagnosis, although these methods also can yield intermediate, 'grey zone' results[36,37]. Regional molecular PET patterns can be an important indicator of observed or expected clinical symptoms[38]. However, recognizing that such evaluations are often not routinely available, a feasible follow-up strategy would be to repeat plasma biomarker testing after, for example, 1 year[12]. The presence of an intermediate zone is an expected and intrinsic feature of continuous biomarker distributions when applied to binary clinical outcomes such as ADNCs. Rather than a limitation, this zone reflects biological and clinical heterogeneity, particularly in population-based cohorts. A substantial proportion of individuals in the intermediate range probably exhibit ADNCs.

The participation rate in HUNT4 70+ was 51.1%, and we are unaware of other similarly large population-based studies with such a high participation rate. Because of limited funding, we were unable to analyse blood samples from all participants 58–69 years of age in HUNT3, but we attempted to adjust for a potential selection bias (see Extended Data Table 9). The cross-sectional design might underestimate amyloid abnormality as opposed to lifetime risk estimates. Furthermore, although nearly 90% of HUNT4 70+ participants provided a blood

sample, 10% did not. The proportion not providing a blood sample was higher in the dementia group, but we adjusted for this in our analysis, as well as for participation bias in HUNT4 (see Extended Data Table 9).

This study has some limitations. Medical diseases were self-reported and thus might not accurately reflect the degree of comorbidities. Previous findings indicate a high NPV and moderate PPV for self-reported diseases in the HUNT study compared with diagnoses recorded in regular clinical care[7]. The HUNT study does not collect data on ethnicity, but the population in this region in 2017 included less than 5% of individuals who are immigrants or Norwegian born to parents who immigrated from Africa, Asia, the Middle East or South America, and thus the findings are relevant for a mainly white Norwegian population. Dementia varies in different ethnic groups and thus the prevalence of ADNCs may also differ in other populations[2].

The predictive value of a diagnostic test varies with the disease prevalence. Thus, as has been demonstrated, when the likelihood of ADNCs is high, for example, in older individuals, the PPV is high, whereas the NPV is lower, that is, there is a risk of false-negative results[30]. This analysis is not a substitute for internal validation and has key limitations: (1) the commutability of sensitivity and/or specificity from external cohorts is assumed; (2) circularity is possible because prevalence is estimated from the same biomarker; and (3) we do not model age-specific shifts in sensitivity and/or specificity or spectrum effects.

In conclusion, we present prevalence estimates of ADNCs in a large, Norwegian population-based cohort. Among individuals 70 years of age or older, 33.4% exhibited ADNCs, with 10% classified as preclinical AD, 10.4% as prodromal AD and 9.8% as AD dementia. Compared with previous studies with smaller, less representative cohorts, our findings indicate a higher prevalence of AD dementia in older individuals and a lower prevalence of preclinical AD in younger age groups.

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

# Methods

## Ethics statement

This study was approved by the Regional Committee for Medical and Health Research Ethics in Norway (REC Southeast C 565876) as well as according to the General Data Protection Regulation by the Norwegian Agency for Shared Services in Education and Research (SIKT 585403). Participation in HUNT required informed consent, which was provided after receiving oral and written information about the health survey. In participants with reduced capacity to consent, their next of kin gave consent. The clinical trial registration number is NCT06719453.

## Cohort selection and study design

The HUNT study is a population-based health study conducted in the Trøndelag region of Central Norway. The key demographic and health indicators of the region mirror Norwegian national averages closely. The HUNT study has so far spanned four waves: HUNT1 (1984–1986), HUNT2 (1995–1997), HUNT3 (2006–2008) and HUNT4 (2017–2019), achieving high participation rates. An updated cohort profile has been previously published[7]. Participants for this study were recruited from HUNT3 and HUNT4.

In HUNT4, all residents 70 years of age or older ($n = 19,463$) were invited to the substudy HUNT4 70+ for a standardized cognitive assessment and diagnosis[11,39] (see below and Supplementary Information). Of the invited, 9,956 people (51.1%) participated, and 8,949 (46%) people provided a blood sample. To also cover younger ages, we additionally included 2,537 out of 12,243 people (20.1%) 58–69.9 years of age from HUNT3. Of these, 2,391 people (94.2%) participated in both HUNT3 and HUNT4 70+. Subsequent cognitive status from HUNT4 70+ and whether they had provided a blood sample for HUNT4 70+ was considered when deciding who to include from HUNT3. We actively selected those from HUNT3 with a diagnosis of dementia in HUNT4 70+ and included twice as many participants with an MCI diagnosis than a cognitively unimpaired status in the subsequent HUNT4 70+. We thereafter accounted for this selection bias in our analysis (see 'Selection bias and weighting'). Thus, the majority of those participants included in the HUNT3 analysis of this study provided two blood biospecimen: one for HUNT3 and one for HUNT4. No cognitive assessment was performed at the time of HUNT3.

Among those not providing a blood sample in HUNT4 70+, there were relatively more from the group with dementia (25.4%) than 8.6% with MCI and 6% from the cognitively unimpaired group. This was also statistically accounted for in our analysis (see 'Selection bias and weighting').

Comorbidities and educational level were self-reported, using a standardized questionnaire.

## Cognitive assessment and diagnosis in HUNT4 70+

Trained health personnel assessed the cognitive, neuropsychiatric and functional status of participants using standardized clinical scales at a field station, at homes or in nursing homes. A structured carer questionnaire was obtained by interview in participants suspected of having substantial cognitive impairment. Clinical and research experts (geriatricians, neurologists and old-age psychiatrists) made diagnoses according to the *Diagnostic and Statistical Manual of Mental Disorders, Fifth Edition* (DSM-5) criteria by clinical consensus method. Participants were categorized into cognitively unimpaired, MCI (minor neurocognitive disorder in the DSM-5) or dementia (major neurocognitive disorder in the DSM-5)[11]. Those with MCI were further classified as amnestic MCI or non-amnestic MCI, whereas dementia was further classified as mild, moderate or severe dementia and according to clinical subtype. The detailed diagnostic procedure has been previously published[11]. A detailed overview over the diagnostic assessment can be found in the Supplementary Information.

## Blood sample collection and handling procedures

Blood samples were collected non-fasting, handled and stored according to standardized procedures[8]. Samples were either collected at field stations, in the homes or in nursing homes of participants. Samples were collected in Vacuette ethylene diamine tetra acetic Acid (EDTA)-plasma 9 ml tubes in HUNT4 and BD Vacutainer EDTA-plasma 10 ml tubes in HUNT3, which were gently inverted (6–8 times after sampling in HUNT4, and 10 times in HUNT3). After a maximum of 45–120 min in room temperature, samples were placed in a refrigerator with a temperature of 2–8 °C before transportation at 4 °C to a central laboratory the same day (until no later than 08:00 the next day in HUNT3). The next day (in HUNT3 on Monday, if blood collection was on a Friday), plasma samples were centrifuged and underwent automated fractioning (Tecan200). Plasma aliquots (tube size of 1.4 ml containing 200 µl plasma) were frozen to −80 °C and stored until transportation for future analyses. Thus, phlebotomy-to-freezing time ensured plasma pTau217 stability[35]. Transportation from the HUNT Biobank to the Clinical Neurochemistry Laboratory, University of Gothenburg (Sweden) was conducted with temperature-regulated dry-ice transport at −80 °C. Samples were further stored at −80 °C until analysis. Before immunoassay analyses, plasma-EDTA samples were thawed, vortexed and centrifuged at 4,000g for 10 min at 20 °C.

## Analysis of plasma pTau217

Using the Simoa HD-X instrument (Quanterix), plasma pTau217 was quantified with a previously validated commercial kit (ALZpath p-Tau 217 Advantage PLUS, Quanterix)[1]. As recommended by the Global CEO Initiative on AD, we used a two cut-off approach to categorize individuals as ADNC negative, intermediate or positive, using recently validated cut-offs: lower cut-off of 0.40 pg ml$^{-1}$ (95% sensitivity) and an upper cut-off of ≥0.63 pg ml$^{-1}$ (95% specificity)[1,12]. We used the upper cut-off to determine with a high specificity those individuals with the presence of ADNC, and the lower cut-off to identify those with a high likelihood of not having ADNC. Plasma analysis was conducted January to August 2024. A summary of the analytical performance of the assay can be found in the Supplementary Information.

## Statistical analyses

**Descriptive analysis.** Descriptive statistics were used to summarize the data. Means and standard deviations were calculated for continuous variables, whereas frequencies were reported for categorical variables. For non-symmetrically distributed data, medians and interquartile ranges were also calculated. All the estimations were weighted for selection bias as explained below. Differences in proportions were assessed using a chi-squared test. Education levels were self-reported and merged into three categories: 'primary' included up to 10 years of compulsory primary and lower secondary education; 'secondary' combined 1–2 years of academic or vocational school, 3 years of academic or vocational school, and 3–4 years of vocational training or apprenticeship (upper secondary education); and 'tertiary' referred to college or university education of less than 4 years or 4 years or more.

**Selection bias and weighting.** To adjust for potential selection bias in participation and data availability, we applied a multi-stage inverse probability weighting strategy. First, we used participation weights developed by Skirbekk et al.[39], which account for differential participation in HUNT4 70+ based on age, sex and educational level. Second, we estimated the probability of donating a blood sample for pTau217 analysis using a logistic regression model that included age, sex, education, *APOE* ε4 status, cognitive diagnosis and self-reported medical disorders. Third, for HUNT3, participants included in this nested study, we modelled the probability of them having been selected among all enrolled HUNT3 participants, based on the cognitive diagnosis established in HUNT4 70+, age, sex, education, *APOE* ε4 status

and self-reported medical disorders. Missing categories in *APOE* ε4 status and education were retained as separate levels in the models.

For each individual, the final weight was calculated as the inverse of the product of the relevant probabilities: three components for participants from HUNT3 (probability of participation in HUNT4, probability of donating a blood sample in HUNT4 70+ and probability of being selected into HUNT3 from HUNT4 70+) and two components for those from HUNT4 70+ (probability of participation in HUNT4 70+ and probability of donating a blood sample in HUNT4 70+). Although covariates were shared across models, each probability addressed a distinct stage of non-random inclusion. Weights were trimmed using the median ± 3 × interquartile range to reduce the influence of outliers[40]. Details of the estimations from the logistic regression models are provided in Extended Data Table 9.

**Imputation of missing covariates for treatment eligibility analyses.** When evaluating the eligible population for novel anti-amyloid immunotherapies in HUNT4 70+ participants with MCI or mild dementia, we performed multiple imputation for missing data for body mass index, *APOE* ε4 status and previous history of stroke or brain haemorrhage, using the mice package in R. The multiple imputation process was conducted using logistic regression for categorical variables (*APOE* ε4 status and stroke or haemorrhage history) and random forest for continuous variables (body mass index), based on a dataset including these and other key demographic variables to inform the imputation models. To estimate the population-representative proportion of eligible individuals, we applied the above-mentioned sampling weights to our eligibility calculations. The final weighted proportion was derived by weighting the sum of all eligible participants from the population (those meeting all clinical and biomarker criteria for either MCI or dementia groups) and dividing them by the weighted sum of the study population. This approach provides an estimate of the true population prevalence of treatment eligibility, accounting for the complex sampling design and differential participation probabilities across the HUNT4 70+ study.

**Model evaluating association between plasma pTau217 and kidney function.** To investigate the relationship between plasma pTau217 concentrations and kidney function, eGFR (ml min$^{-1}$ per 1.73 m$^2$) was calculated using the CKD-EPI 2021 creatinine-based equation[41]. Plasma pTau217 was log-transformed, and a weighted linear regression model was fitted with eGFR as a predictor. A piecewise (segmented) regression approach was then applied to allow different slopes on either side of an estimated breakpoint in eGFR. The final model was visualized on the original (non-logarithmic) scale, showing the fitted regression lines for each segment and indicating the estimated inflection point.

**Predictive value sensitivity analysis.** In an exploratory analysis, we estimated age-stratum-specific PPV and NPV for plasma pTau217, with externally derived sensitivity and specificity ranges for the ability of plasma pTau217 to detect ADNC as reported by Ashton et al.[1]. For each age stratum (70–74, 75–79, 80–84, 85–89, 90 and older, and overall), prevalence was set to the weighted prevalence of plasma pTau217 positivity (≥0.63 pg ml$^{-1}$) from HUNT4 70+, using inverse-probability weights reflecting participation and selection into the plasma pTau217 subsample. We enumerated all combinations of sensitivity (0.850, 0.982) and specificity (0.745, 0.986) in 0.001 increments and, for each triplet (sensitivity (Se), specificity (Sp) and prevalence (Prev)), computed PPV and NPV via Bayes' formulas: PPV = (Se × Prev)/(Se × Prev + (1 − Sp) × (1 − Prev)); NPV = (Sp × (1 − Prev))/((1 − Se) × Prev + Sp × (1 − Prev)). For each age stratum, we summarized the empirical distributions with the median and the 2.5th–97.5th percentiles (reported as 95% confidence interval). Given this exploratory analysis assumes sensitivity and specificity commutability from Ashton et al.[1] and that a certain degree of worsened performance is generally expected upon external validation studies, we repeated the analyses above at a sensitivity and specificity shrinkage level of 0.9, to correct for optimism.

## Reporting summary

Further information on research design is available in the Nature Portfolio Reporting Summary linked to this article.

## Data availability

To protect the privacy of participants, the HUNT Research Centre aims to limit storage of data outside the HUNT databank and cannot deposit data in open repositories. The HUNT databank has precise information on all data exported to different projects and can reproduce them on request. There are no restrictions regarding data export given approval of applications to the HUNT Research Centre. Researchers can apply for data at the NTNU (https://www.ntnu.edu/hunt/research).

## Code availability

All analyses, statistical models and figures were performed using the open-source statistical software environment and programming language R (v4.5.0; R Core Team). All codes for data processing and analysis are publicly available on GitHub (https://github.com/dalejo643/Nature-Aarsland2025.git). All models relied on publicly available R packages and functions.

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

**Acknowledgements** The study was funded by the Norwegian Helse Vest trust (F-12850 and F-13067). The HUNT study is a collaboration between the HUNT Research Centre (Faculty of Medicine and Health Sciences, Norwegian University of Science and Technology NTNU), Trøndelag County Council, Central Norway Regional Health Authority and the Norwegian Institute of Public Health.

**Author contributions** D.A., A.L.S., D.A.T.R., H.K.S., G.S. and N.J.A. designed the study. D.A., A.L.S., D.A.T.R., T.F., H.K.S. and G.S. participated in data collection and/or management. H.Z., K.B., K.T., G.D.S., Y.Y., B.A., H.H., I.P., L.G., G.D.M. and N.J.A. coordinated and/or performed the blood biomarker quantification. D.A.T.R. and A.L. performed the data analyses. D.A., A.L.S., D.A.T.R. and A.L. wrote the initial draft of the manuscript. All authors contributed to the interpretation of the results and towards subsequent manuscript drafts.

**Competing interests** D.A. has received research support and/or honoraria from AstraZeneca, H. Lundbeck, Novartis Pharmaceuticals, Evonik, Roche Diagnostics, GE Health, Bioarctic and Sanofi; and served as a paid consultant for H. Lundbeck, Eisai, Heptares, Mentis Cura, Eli Lilly, Cognetivity, Enterin, Acadia, EIP Pharma, Biogen and Takeda. A.L. has acted as a consultant for Enigma Biomedical Group. H.Z. has served on scientific advisory boards and/or as a consultant for AbbVie, Acumen, Alector, Alzinova, ALZpath, Amylyx, Annexon, Apellis, Artery Therapeutics, AZTherapies, Cognito Therapeutics, CogRx, Denali, Eisai, Enigma, LabCorp, Merry Life, Nervgen, Novo Nordisk, Optoceutics, Passage Bio, Pinteon Therapeutics, Prothena, Quanterix, Red Abbey Labs, reMYND, Roche, Samumed, Siemens Healthineers, Triplet Therapeutics and Wave; has given lectures sponsored by Alzecure, BioArctic, Biogen, Cellectricon, Fujirebio, Lilly, Novo Nordisk, Roche and WebMD; and is a co-founder of Brain Biomarker Solutions in Gothenburg AB (BBS), which is a part of the GU Ventures Incubator Program (outside the submitted work). K.B. has served as a consultant and on advisory boards for AbbVie, AC Immune, ALZPath, AriBio, Beckman-Coulter, BioArctic, Biogen, Eisai, Lilly, Moleac, Neurimmune, Novartis, Ono Pharma, Prothena, Quanterix, Roche Diagnostics, Sanofi and Siemens Healthineers; has served on data monitoring committees for Julius Clinical and Novartis; has given lectures, produced educational materials and participated in educational programs for AC Immune, Biogen, Celdara Medical, Eisai and Roche Diagnostics; and is a co-founder of Brain Biomarker Solutions in Gothenburg AB (BBS), which is a part of the GU Ventures Incubator Program, outside the work presented in this paper. G.S. has participated in advisory board meetings for Roche, Eli-Lilly and Eisai regarding disease-modifying drugs for AD; and has received honoraria for delivering lectures at symposia sponsored by Eisai and Eli-Lilly. N.J.A. has given lectures, produced educational materials and participated in educational programs for Eli-Lily, BioArtic and Quanterix. A.L.S., D.A.T.R., T.F., K.T., G.D.S., L.G., Y.Y., B.A., H.H., I.P., G.D.M. and H.K.S. declare no competing interests.

**Additional information**
**Correspondence and requests for materials** should be addressed to Dag Aarsland or Anita Lenora Sunde.

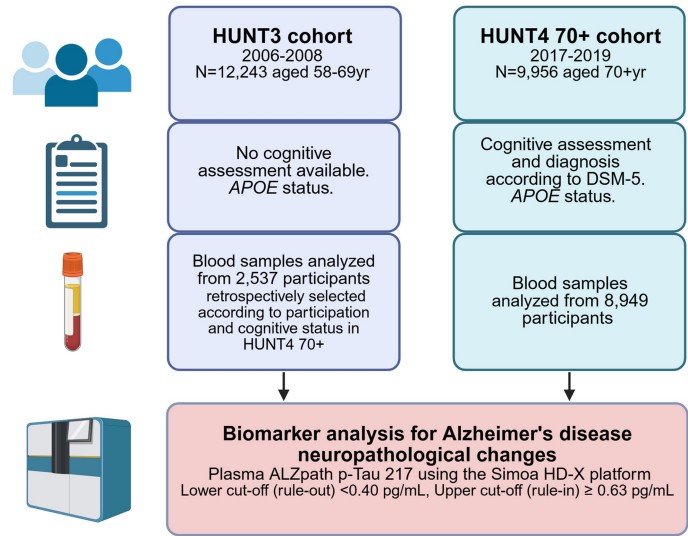

**Extended Data Fig. 1 | Study overview.** HUNT = The Trøndelag Health Study, Wave 3 and 4; yr = years; DSM-5 = The Diagnostic and Statistical Manual of Mental Disorders, Fifth Edition; *APOE* = apolipoprotein E; p-Tau 217 = tau phosphorylated at threonine 217; pg/mL = picograms per millilitre. Plasma pTau217 was measured with ALZpath p-Tau 217 Advantage PLUS, Quanterix. The figure was created using BioRender (https://biorender.com).

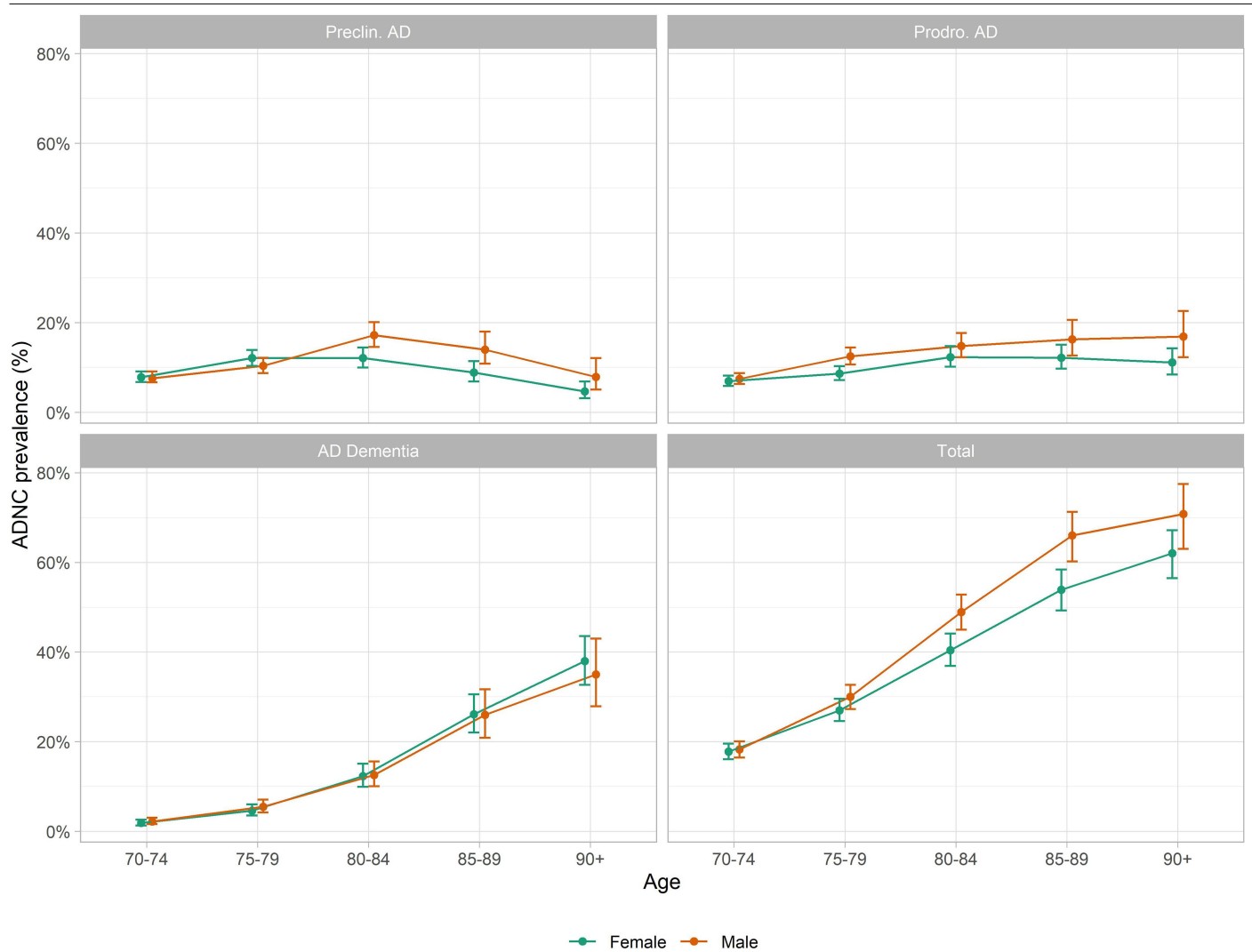

**Extended Data Fig. 2 | Association between the prevalence of ADNC, age and sex in different cognitive groups.** ADNC = Alzheimer's disease neuropathological changes, defined as plasma pTau217 ≥ 0.63 picograms per millilitre; Preclin = Preclinical; Prodro = Prodromal. N = 8,949 participants from HUNT4 70+. Dots represent the weight-estimated ADNC prevalence for each sex in 5-year age groups, shown for each cognitive group and for the total. The whiskers are 95% confidence intervals for each estimation.

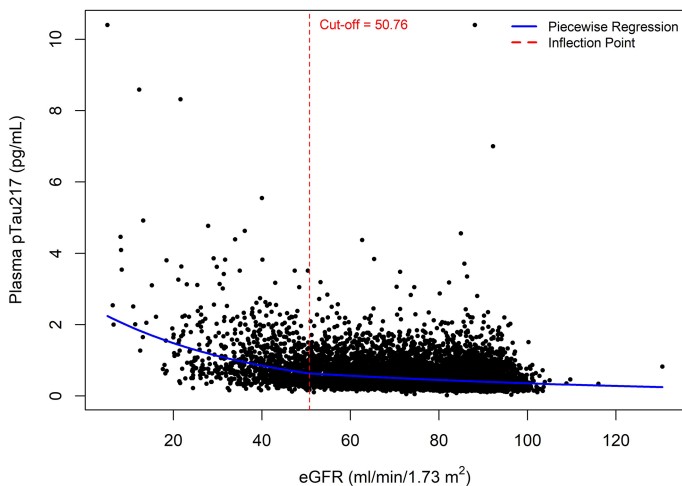

**Extended Data Fig. 3 | Piecewise regression analysis of plasma pTau217 and eGFR, showing fitted regression lines and identified cut-off point.** pTau217 = tau phosphorylated at threonine 217; pg/mL = picograms per millilitre; eGFR = estimated glomerular filtration rate. N = 8942 participants from HUNT4 70+. Black dots represent individual data points of plasma pTau217 concentrations and corresponding estimated glomerular filtration rates. The blue solid line represents the fitted piecewise regression model, which allows for different slopes on either side of a threshold. The red dashed vertical line indicates the inflection point (cutoff) at eGFR 50.76 ml/min/1.73 m², where the slope of the regression changes.

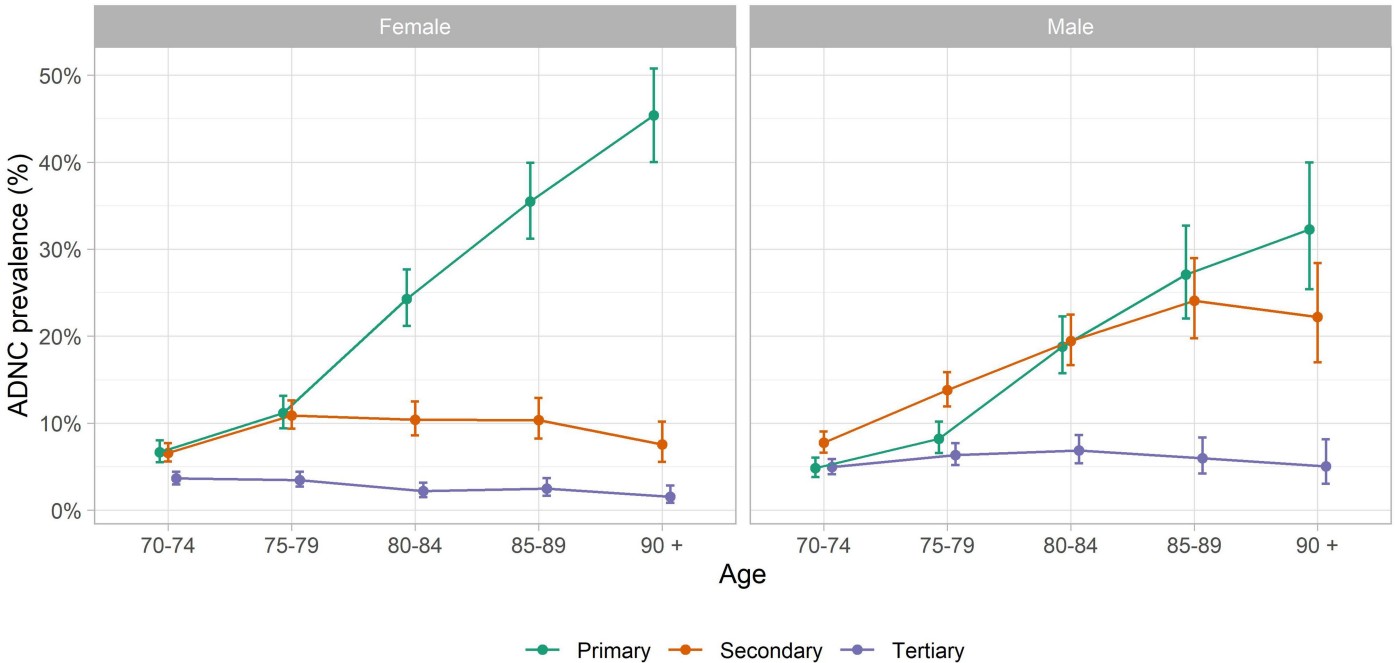

**Extended Data Fig. 4 | Estimated prevalence of ADNC across age and educational groups.** ADNC = Alzheimer's disease neuropathological changes, defined as plasma pTau217 ≥ 0.63 picograms per millilitre; Primary Education = up to 10 years of compulsory primary and lower secondary education; Secondary Education = 1–2 years of academic or vocational school, 3 years of academic or vocational school, or 3–4 years of vocational training or apprenticeship; Tertiary Education = college or university education of less than four years or four years or more. N = 8949 participants from HUNT4 70+. Dots represent the weight-estimated ADNC prevalence for each sex in every age group for every education level. The whiskers are 95% confidence intervals for each estimation.

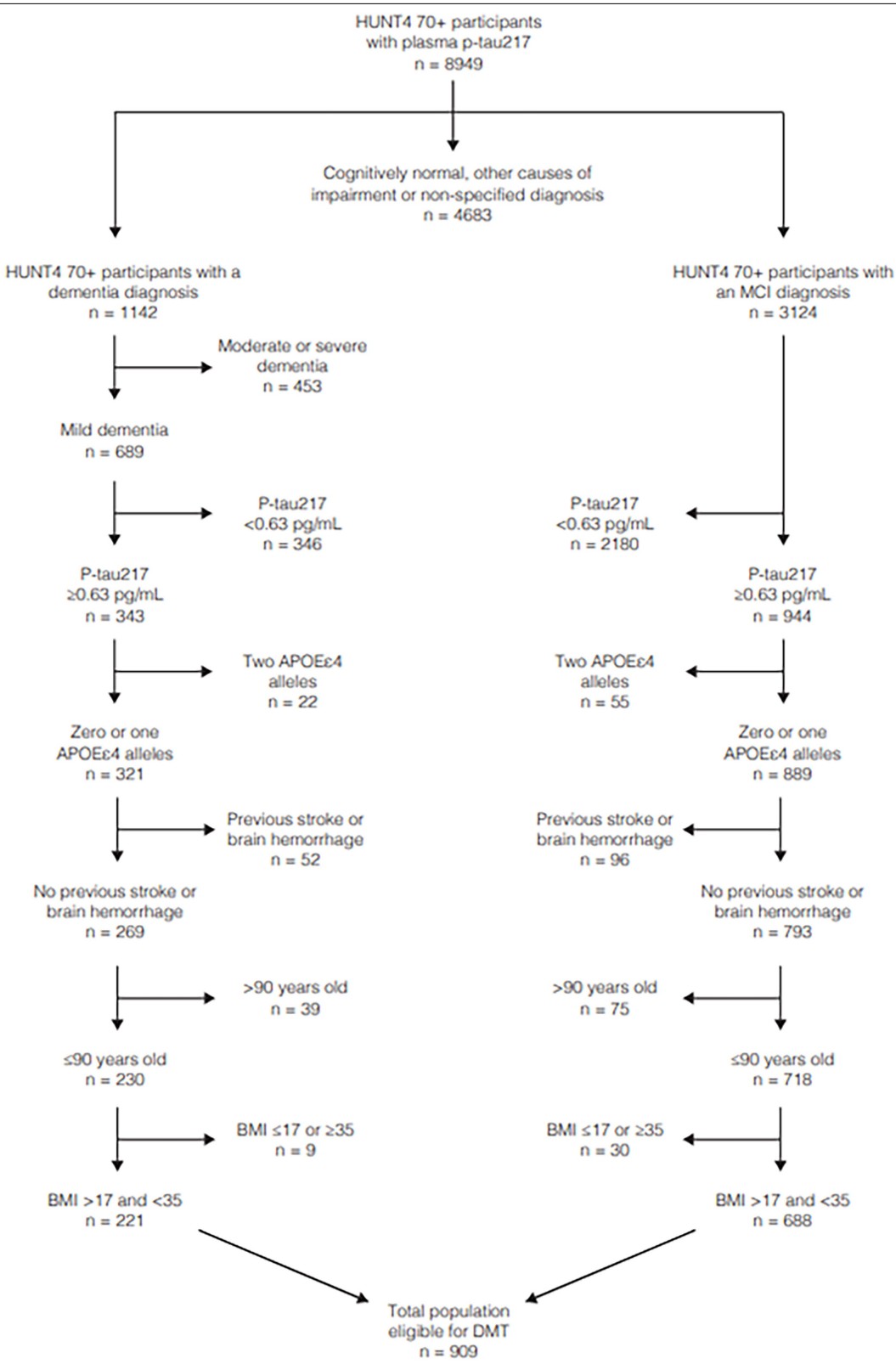

**Extended Data Fig. 5 | Algorithm for identifying people eligible for treatment with disease-modifying drugs against Alzheimer's disease according to recommendations.** HUNT = The Trøndelag Health Study; MCI = mild cognitive impairment; pTau217 = tau phosphorylated at threonine 217; pg/mL = picograms per millilitre; *APOE* = apolipoprotein E; BMI = Body Mass Index; DMT = disease-modifying treatment.

**Extended Data Table 1 | Demographic and clinical characteristics of the cohort**

|  | HUNT 3 | HUNT 4 70+ |
|---|---|---|
| Total, n | 2537 | 8949 |
| Age, yr |  |  |
| Mean, sd | 64.0 ± 2.98 | 77.6 ± 6.22 |
| Median, min, max | 63.7 [58.0; 69.9] | 76 [70; 100] |
| Female, n (%) | 1298 (51.2) | 4798 (53.6) |
| ApoE ε4 carrier, n (%) |  |  |
| ε3ε3/ε3ε2/ε2ε2 | 1698 (66.9) | 6251 (68.9) |
| ε2ε4/ε3ε4 | 723 (28.5) | 2405 (26.9) |
| ε4ε4 | 97 (3.82) | 234 (2.6) |
| Missing | 19 (0.75) | 59 (0.66) |
| Education, n (%) |  |  |
| Primary | 717 (28.3) | 2755 (30.8) |
| Secondary | 1182 (46.6) | 3902 (43.6) |
| Tertiary | 632 (24.9) | 2232 (24.9) |
| Missing | 6 (0.24) | 60 (0.67) |

HUNT=The Trøndelag Health Study, Wave 3 and 4; yr=years; sd=standard deviation; *APOE*=apolipoprotein E. "Primary education" included up to 10 years of compulsory primary and lower secondary education; "Secondary education" was defined as 1–2 years of academic or vocational school, 3 years of academic or vocational school, or 3–4 years of vocational training or apprenticeship (upper secondary education); "Tertiary education" referred to college or university education of less than four years or four years or more.

**Extended Data Table 2 | Estimated age dependent proportion of plasma pTau217 concentration below the lower cut-off (rule-out) for ADNC, above the upper cut-off (rule-in) and in the intermediate group**

| Age group (yr) | n | pTau217 mean ± se | Cut-off groups (pg/mL) < 0.40 | 0.40 – 0.63 | ≥ 0.63 |
|---|---|---|---|---|---|
| 58 - 59 | 174 | 0.30 ± 0.014 | 140 (82.2) [76.1 - 87.3] | 24 (13.9) [9.33 - 19.6] | 10 (3.91) [1.82 - 7.01] |
| 60 - 64 | 1400 | 0.29 ± 0.005 | 1144 (84.1) [82.1 - 85.8] | 214 (13.5) [11.9 - 15.3] | 42 (2.47) [1.80 - 3.28] |
| 65 - 69 | 963 | 0.35 ± 0.007 | 689 (74.4) [71.7 - 76.9] | 181 (18.0) [15.7 - 20.3] | 93 (7.69) [6.28 - 9.29] |
| 70 - 74 | 3828 | 0.45 ± 0.006 | 2221 (57.7) [56.1 - 59.3] | 934 (24.3) [23.0 - 25.7] | 673 (18.0) [16.8 - 19.3] |
| 75 - 79 | 2446 | 0.55 ± 0.008 | 1089 (44.0) [42.0 - 46.0] | 673 (27.6) [25.8 - 29.4] | 684 (28.3) [26.6 - 30.2] |
| 80 - 84 | 1416 | 0.72 ± 0.016 | 415 (28.5) [26.2 - 31.0] | 396 (27.3) [25.0 - 29.7] | 605 (44.1) [41.6 - 46.9] |
| 85 - 89 | 772 | 0.91 ± 0.029 | 127 (16.) [13.6 - 18.8] | 206 (25.7) [22.7 - 28.9] | 439 (57.9) [54.7 - 61.7] |
| 90 + | 487 | 0.96 ± 0.030 | 77 (15.6) [12.6 - 19.1] | 94 (19.7) [16.2 - 23.4] | 316 (65.2) [60.3 - 68.9] |

pTau217 = tau phosphorylated at threonine 217; yr = years; se = Linearized standard error; pg/mL = picograms per millilitre. Numbers in brackets represent 95% confidence intervals. The estimated prevalences for individuals aged 58 to 69.9 years were calculated using weights that account for the probability of selection from all HUNT3 participants. The prevalences for individuals aged 70+ years were calculated using weights that account for the probability of both participating and providing a blood sample in HUNT4 70+. These weights correct for selection bias and ensure representativeness of the target population.

**Extended Data Table 3 | Estimated proportions with and without ADNC in people 70+ years with dementia, mild cognitive impairment or cognitively unimpaired**

| Age group (yr) | n (%) pTau217 < 0.40 pg/mL | | | | P-value | n (%) pTau217 ≥ 0.63 pg/mL | | | | P-value |
| --- | --- | --- | --- | --- | --- | --- | --- | --- | --- | --- |
| | CU | MCI | Dementia | Total | | CU | MCI | Dementia | Total | |
| 70 - 74 | 1363 (61.6) [59.5 - 63.6] | 755 (54.8) [52.2 - 57.5] | 83 (41.0) [34.3 - 47.9] | 2221 (57.7) [56.1 - 59.3] | <.001 | 319 (14.6) [13.2 - 16.2] | 274 (20.3) [18.1 - 22.5] | 71 (33.1) [26.8 - 39.8] | 673 (18.0) [16.8 - 19.3] | <.001 |
| 75 - 79 | 643 (48.7) [46.0 - 51.5] | 386 (43.4) [40.1 - 46.7] | 50 (23.7) [18.2 - 29.8] | 1089 (44.0) [42.0 - 46.0] | <.001 | 302 (22.9) [20.7 - 25.3] | 264 (29.9) [26.9 - 33.0] | 108 (49.6) [42.8 - 56.4] | 684 (28.4) [26.6 - 30.2] | <.001 |
| 80 - 84 | 244 (35.0) [31.4 - 38.7] | 130 (27.9) [23.9 - 32.2] | 39 (16.2) [11.8 - 21.4] | 415 (28.5) [26.2 - 31.0] | <.001 | 240 (34.8) [31.2 - 38.5] | 206 (45.1) [40.5 - 49.8] | 149 (63.5) [57.1 - 69.6] | 605 (44.2) [41.6 - 46.9] | <.001 |
| 85 - 89 | 58 (24.6) [19.2 - 30.3] | 31 (15.3) [11.1 - 20.4] | 29 (11.5) [7.89 - 16.0] | 127 (16.1) [13.6 - 18.8] | 0.005 | 116 (45.7) [39.4 - 52.1] | 134 (53.7) [47.3 - 60.0] | 169 (72.0) [66.0 - 77.5] | 439 (58.2) [54.7 - 61.7] | <.001 |
| 90 + | 19 (24.7) [15.9 - 36.2] | 17 (12.1) [7.32 - 18.2] | 35 (14.7) [10.5-19.7] | 77 (15.6) [12.6 - 19.1] | 0.284 | 45 (62.5) [50.6 - 73.5] | 89 (60.7) [52.4 - 68.6] | 162 (66.7) [60.5 - 72.6] | 316 (64.7) [60.3 - 68.9] | 1.000 |
| All | 2327 (50.1) [48.6 - 51.6] | 1325 (41.0) [39.2 - 42.7] | 236 (19.4) [17.2 - 21.7] | 3929 (41.1) [40.1 - 42.2] | <.001 | 1022 (23.5) [22.2 - 24.8] | 967 (32.6) [30.9 - 34.3] | 659 (60.0) [57.0 - 62.8] | 2717 (33.4) [32.4 - 34.4] | <.001 |

pTau217 = tau phosphorylated at threonine 217; pg/mL = picograms per millilitre; CU=cognitively unimpaired; MCI=mild cognitive impairment; yr=years. Numbers in brackets represent 95% confidence intervals. All p-values were adjusted for multiple testing with the Bonferroni method. The proportions were calculated by dividing the number of cases in each cognitive and age group (classified as below or above the cut-off) by the total number of participants in that cognitive group, weighted by the inversed probability of participating and providing a blood sample in HUNT4 70+.

**Extended Data Table 4 | Weighted estimated proportions of plasma pTau217 categories (ADNC negative, intermediate, positive) by cognitive subgroups in people 70+ years**

| Cognitive subgroup | N | n (%) pTau217 < 0.40 pg/mL | n (%) 0.40 ≤ pTau217 < 0.63 pg/mL | n (%) pTau217 ≥ 0.63 pg/mL |
|---|---|---|---|---|
| aMCI | 2,667 | 1,134 (41.3) [39.4 - 43.2] | 694 (25.6) [23.9 - 27.3] | 839 (33.1) [31.3 - 35.0] |
| naMCI | 457 | 191 (39.7) [35.1 - 44.2] | 138 (31.3) [26.9 - 35.7] | 128 (29.0) [24.7 - 33.3] |
| ADD | 667 | 125 (17.5) [14.6 - 20.4] | 139 (20.1) [17.0 - 23.2] | 403 (62.4) [58.7 - 66.2] |
| VaD | 118 | 34 (30.8) [22.0 - 39.6] | 31 (24.6) [16.7 - 32.6] | 53 (44.6) [35.3 - 53.9] |
| DLB/PDD | 37 | 9 (24.7) [10.4 - 38.9] | 9 (25.4) [10.9 - 40.0] | 19 (49.9) [33.4 - 66.4] |
| FTD | 30 | 8 (22.1) [7.70 - 36.6] | 8 (25.4) [9.60 - 41.3] | 14 (52.5) [34.0 - 70.9] |
| Mixed dementia | 99 | 11 (11.3) [4.80 - 17.8] | 20 (20.2) [12.0 - 28.3] | 68 (68.5) [59.1 - 78.0] |
| Unspecified dementia | 189 | 47 (23.0) [17.0 - 29.0] | 40 (20.1) [14.3 - 25.9] | 102 (56.9) [49.7 - 64.2] |

pTau217=tau phosphorylated at threonine 217; ADNC=Alzheimer's disease neuropathological changes; pg/mL=picograms per millilitre; aMCI=amnestic mild cognitive impairment; naMCI=non-amnestic mild cognitive impairment; ADD=Alzheimer's disease dementia; VaD=vascular dementia; DLB=dementia with Lewy bodies; PDD=Parkinson's disease dementia; FTD=frontotemporal dementia.

**Extended Data Table 5 | Estimated prevalence of preclinical AD, prodromal AD and AD dementia**

| Age group (yr) | | Female, n (%) | | | | Male, n (%) | | | | Total, n (%) | | |
|---|---|---|---|---|---|---|---|---|---|---|---|---|
| | N | Preclin AD | Prodro AD | AD Dementia | N | Preclin AD | Prodro AD | AD Dementia | N | Preclin AD | Prodro AD | AD Dementia |
| 70 - 74 | 1947 | 164 (7.85) [6.75 - 9.11] | 134 (6.97) [5.90 - 8.22] | 32 (1.85) [1.30 - 2.62] | 1881 | 155 (7.86) [6.73 - 9.16] | 140 (7.46) [6.33 - 8.77] | 39 (2.23) [1.62 - 3.07] | 3828 | 319 (7.85) [7.08 - 8.74] | 274 (7.21) [6.41 - 8.09] | 71 (2.03) [1.61 - 2.57] |
| 75 - 79 | 1286 | 169 (12.1) [10.4 - 13.9] | 116 (8.63) [7.22 - 10.3] | 52 (4.43) [3.38 - 5.78] | 1160 | 133 (10.4) [8.77 - 12.2] | 148 (12.5) [10.7 - 14.5] | 56 (5.49) [4.23 - 7.09] | 2446 | 302 (11.3) [10.1 - 12.5] | 264 (10.4) [9.26 - 11.7] | 108 (4.92) [4.08 - 5.92] |
| 80 - 84 | 760 | 107 (12.1) [10.0 - 14.5] | 103 (12.3) [10.2 - 14.8] | 78 (11.5) [9.30 - 14.2] | 656 | 133 (17.2) [14.6 - 20.1] | 103 (14.8) [12.3 - 17.7] | 71 (12.6) [10.1 - 15.6] | 1416 | 240 (14.3) [12.7 - 16.1] | 206 (13.4) [11.8 - 15.2] | 149 (12.0) [10.3 - 13.9] |
| 85 - 89 | 480 | 59 (8.87) [6.88 - 11.4] | 75 (12.2) [9.74 - 15.1] | 105 (24.9) [21.0 - 29.1] | 292 | 57 (14.0) [10.9 - 18.0] | 59 (16.3) [12.7 - 20.6] | 64 (26.0) [20.9 - 31.7] | 772 | 116 (10.7) [8.95 - 12.8] | 134 (13.6) [11.6 - 16.0] | 169 (25.2) [22.1 - 28.6] |
| 90 + | 325 | 25 (4.68) [3.15 - 6.90] | 51 (11.1) [8.44 - 14.3] | 110 (35.1) [30.0 - 40.4] | 162 | 20 (7.92) [5.08 - 12.1] | 38 (16.9) [12.33 - 22.6] | 52 (35.0) [27.9 - 43.0] | 487 | 45 (5.64) [4.21 - 7.54] | 89 (12.8) [14.4 - 15.6] | 162 (35.1) [30.9 - 39.5] |
| All | 4798 | 524 (9.35) [8.59 - 10.2] | 479 (9.49) [8.69 -10.4] | 377 (10.5) [9.51 - 11.5] | 4151 | 498 (10.8) [9.90 - 11.7] | 488 (11.5) [10.5 - 12.5] | 282 (8.82) [7.87 - 9.88] | 8949 | 1022 (9.98) [9.40 - 10.6] | 967 (10.4) [9.75 - 11.0] | 659 (9.75) [9.05 - 10.5] |

AD=Alzheimer's disease; Preclin=Preclinical; Prodro=Prodromal. Preclinical AD is defined as cognitively unimpaired people (CU) having Alzheimer's disease neuropathological changes. Prodromal AD is defined as people with mild cognitive impairment (MCI) having Alzheimer's disease neuropathological changes. AD dementia is defined as people with dementia having Alzheimer's disease neuropathological changes. Prevalences were calculated by dividing the number of cases in each cognitive and age group above the plasma pTau217 cut-off (≥0.63 picograms per millilitre) by the total number of participants in each age group, weighted by the inversed probability of participating and providing a blood sample in HUNT4 70+, separately for each sex and overall. Numbers in brackets represent 95% confidence intervals. Of the 8,949 participants, 153 could not be categorized as either CU, MCI or dementia due to missing information. Of those, 69 (0.77% of the total) had ANDC (pTau217 ≥0.63 picograms per millilitre).

**Extended Data Table 6 | Estimated proportions of ADNC in people 70+ years by *APOE* genotype and cognitive status**

| pTau217 | ApoE ε4 allele | CU | MCI | Dementia | Total | P-value |
|---|---|---|---|---|---|---|
| < 0.40 | ε3ε3/ε3ε2/ε2ε2 | 1897 (56.5) [54.7 - 58.2] | 1075 (47.9) [45.8 - 50.1] | 200 (26.5) [19.9 - 36.3] | 3205 (48.3) [47.1 - 49.6] | <.001 |
| | ε2ε4/ε3ε4 | 405 (34.1) [31.4 - 36.9] | 232 (26.6) [23.6 - 29.6] | 30 (8.35) [3.57 - 15.8] | 675 (25.8) [24.1 - 27.6] | <.001 |
| | ε4ε4 | 18 (26.6) [23.4 - 29.9] | 7 (7.46) [5.10 - 10.4] | 3 (6.56) [1.61 - 16.4] | 28 (11.9) [8.11 - 16.6] | 0.206 |
| 0.40 – 0.63 | ε3ε3/ε3ε2/ε2ε2 | 770 (23.9) [22.4 - 25.4] | 539 (25.0) [23.2 - 26.9] | 182 (24.8) [21.6 - 28.1] | 1526 (24.6) [23.5 - 25.7] | 1.000 |
| | ε2ε4/ε3ε4 | 381 (33.1) [30.4 - 35.9] | 260 (30.1) [27.0 - 33.3] | 54 (13.7) [10.4 - 17.5] | 701 (27.8) [26.0 - 29.6] | <.001 |
| | ε4ε4 | 26 (30.2) [20.9 - 40.8] | 22 (24.6) [16.2 - 34.5] | 8 (14.1) [6.54 - 25.0] | 58 (23.5) [18.3 - 29.2] | 0.946 |
| ≥ 0.63 | ε3ε3/ε3ε2/ε2ε2 | 603 (19.6) [18.2 - 21.1] | 541 (27.1) [25.1 - 29.1] | 332 (48.7) [44.9 - 52.4] | 1520 (27.1) [25.9 - 28.3] | <.001 |
| | ε2ε4/ε3ε4 | 375 (32.8) [30.1 - 35.6] | 358 (43.3) [39.9 - 46.8] | 275 (78.9) [74.5 - 82.9] | 1029 (46.4) [44.4 - 48.5] | <.001 |
| | ε4ε4 | 41 (48.3) [37.5 - 59.3] | 58 (67.1) [56.5 - 76.6] | 46 (79.4) [66.9 - 88.9] | 148 (64.6) [58.2 - 70.6] | 0.013 |

pTau217 = tau phosphorylated at threonine 217, measured in picograms per millilitre; *APOE* = apolipoprotein E; CU=cognitively unimpaired; MCI=mild cognitive impairment. All p-values were adjusted for multiple testing with the Bonferroni method. Proportions were calculated by dividing the number of participants from each pTau217 category (ADNC negative, intermediate, positive) by the total number of participants within each combination of cognitive diagnosis and *APOE* ε4 allele count and were weighted by the inverse probability of participating and providing a blood sample in HUNT4 70+.

**Extended Data Table 7 | Association between self-reported medical disorders and ADNC**

| Comorbidity | N | n (%) | mean ± se, pTau217, pg/mL | | n (%) pTau217 ≥ 0.63 pg/mL | | OR | 95% CI | P-value | Adj. P-value |
|---|---|---|---|---|---|---|---|---|---|---|
| | | | No | Yes | No | Yes | | | | |
| Angina pectoris | 6987 | 853 (13.7) | 0.56 ± 0.007 | 0.72 ± 0.021 | 1660 (29.2) | 337 (42.6) | 0.99 | [0.82 - 1.20] | 0.937 | 1.000 |
| Myocardial infarction | 7056 | 936 (14.5) | 0.56 ± 0.007 | 0.69 ± 0.021 | 1665 (29.4) | 356 (40.7) | 1.08 | [0.90 - 1.30] | 0.395 | 1.000 |
| Heart failure | 6900 | 506 (8.78) | 0.56 ± 0.007 | 0.81 ± 0.029 | 1747 (29.6) | 247 (51.0) | 1.05 | [0.83 - 1.34] | 0.670 | 1.000 |
| Atrial fibrillation° | 7672 | 1081 (14.3) | 0.57 ±0.007 | 0.62 ± 0.014 | 1796 (29.6) | 364 (36.5) | 1.07 | [0.91 - 1.27] | 0.388 | 1.000 |
| Stroke/brain haemorrhage | 6992 | 841 (13.8) | 0.57 ± 0.006 | 0.66 ± 0.028 | 1714 (30.2) | 286 (35.9) | 0.86 | [0.71 - 1.05] | 0.137 | 1.000 |
| COPD or emphysema | 7034 | 766 (11.9) | 0.58 ± 0.007 | 0.60 ± 0.017 | 1772 (30.7) | 245 (33.9) | 1.00 | [0.82 - 1.22] | 0.973 | 1.000 |
| Diabetes | 7475 | 1029 (14.8) | 0.57 ± 0.006 | 0.67 ± 0.024 | 1807 (30.4) | 354 (37.1) | 1.01 | [0.85 - 1.20] | 0.935 | 1.000 |
| Cancer | 7232 | 1681 (24.0) | 0.58 ± 0.008 | 0.62 ± 0.014 | 1542 (30.0) | 535 (34.5) | 1.03 | [0.89 - 1.18] | 0.726 | 1.000 |
| Migraine° | 7813 | 963 (11.7) | 0.59 ± 0.007 | 0.58 ± 0.016 | 1971 (31.1) | 257 (30.1) | 0.99 | [0.83 - 1.19] | 0.916 | 1.000 |
| Psoriasis | 7005 | 723 (10.5) | 0.58 ± 0.007 | 0.59 ± 0.022 | 1728 (30.8) | 210 (30.8) | 0.95 | [0.78 - 1.17] | 0.658 | 1.000 |
| Kidney disease (other than urinary tract infection) | 6970 | 588 (9.05) | 0.56 ± 0.006 | 0.87 ± 0.049 | 1746 (29.7) | 257 (46.8) | 1.16 | [0.93 - 1.44] | 0.200 | 1.000 |
| Rheumatoid arthritis° | 7730 | 783 (10.5) | 0.58 ± 0.006 | 0.60 ± 0.018 | 1964 (30.7) | 230 (31.9) | 1.01 | [0.83 - 1.22] | 0.952 | 1.000 |
| Spondyloarthritis° | 7687 | 99 (1.30) | 0.58 ± 0.007 | 0.58 ± 0.047 | 2154 (30.8) | 29 (32.0) | 1.05 | [0.65 - 1.68] | 0.841 | 1.000 |
| Gout° | 7737 | 691 (9.31) | 0.57 ± 0.006 | 0.71 ± 0.024 | 1955 (30.1) | 252 (40.0) | 1.00 | [0.82 - 1.22] | 0.974 | 1.000 |
| Seeking help for mental health problem° | 7792 | 939 (12.5) | 0.58 ± 0.007 | 0.59 ± 0.018 | 1964 (31.0) | 260 (30.8) | 0.90 | [0.75 - 1.09] | 0.285 | 1.000 |
| Epylepsy* | 2442 | 42 (1.54) | 0.47 ± 0.005 | 0.52 ± 0.051 | 608 (20.5) | 9 (26.3) | 1.44 | [0.44 - 4.72] | 0.549 | 1.000 |
| Eczema on hands* | 2537 | 245 (9.64) | 0.48 ± 0.005 | 0.45 ± 0.012 | 581 (20.8) | 66 (19.7) | 1.22 | [0.68 - 2.20] | 0.510 | 1.000 |

* Assessed only in HUNT3 aged 58 – 69 years. ¤ Assessed only in HUNT4 70+. pTau217 = tau phosphorylated at threonine 217, measured in picograms per millilitre (pg/mL); se=Linearized standard error. Alzheimer's disease neuropathological changes (ADNC) are defined as plasma pTau217 concentration ≥ 0.63 pg/mL. Adjusted p-values (Bonferroni correction) from separate logistic regression models predicting ADNC positivity (pTau217 ≥ 0.63 pg/mL vs pTau217 < 0.63 pg/mL), each adjusted for age (1-year intervals), sex, APOE ε4 allele count, cognition, serum creatinine, and education level. All models were re-run after excluding participants in the intermediate range (0.40–0.63 pg/mL), redefining the outcome as pTau217 ≥ 0.63 pg/mL vs. pTau217 < 0.40 pg/mL. The results remained unchanged, with no significant differences in ADNC positivity between participants with and without the self-reported medical condition for any of the disorders assessed.

**Extended Data Table 8 | Positive Predictive Value and Negative Predictive Value sensitivity analysis by age group**

| Age group (yr) | Prevalence | PPV | | NPV | |
|---|---|---|---|---|---|
| | | Unpenalized | Optimism-corrected | Unpenalized | Optimism-corrected |
| 70 - 74 | 18.0 | 59.9 [44.3 - 91.1] | 52.9 [40.8 - 74.0] | 97.9 [96.2 - 99.5] | 96.8 [95.1 - 98.3] |
| 75 - 79 | 28.4 | 73.0 [59.0 - 94.9] | 67.0 [55.5 - 83.7] | 96.3 [93.4 - 99.1] | 94.3 [91.5 - 96.9] |
| 80 - 84 | 44.2 | 84.4 [74.2 - 97.4] | 80.2 [71.3 - 91.1] | 92.9 [87.6 - 98.1] | 89.3 [84.4 - 94.0] |
| 85 - 89 | 58.2 | 90.5 [83.5 - 98.5] | 87.7 [81.4 - 94.7] | 88.1 [80.0 - 96.8] | 82.6 [75.5 - 89.9] |
| 90 + | 64.7 | 92.6 [86.9 - 98.8] | 90.4 [85.2 - 96.0] | 84.9 [75.3 - 95.8] | 78.3 [70.0 - 87.1] |
| All | 33.4 | 77.4 [64.5 - 95.9] | 71.9 [61.1 - 86.6] | 95.4 [91.7 - 98.8] | 92.9 [85.5 - 96.1] |

yr = year; PPV = Positive Predictive Value; NPV = Negative Predictive Value; Se=Sensitivity; Sp=Specificity. Unpenalized analysis: Se: 0.850–0.982; Sp: 0.745–0.986; medians with 2.5–97.5% quantiles. Optimism-corrected: Se/Sp shrunk 10% toward 0.5 (Se* = 0.5+0.9·(Se–0.5); Sp* analogous).

**Extended Data Table 9 | Logistic regression models used to estimate probabilities for inverse probability weighting: selection into HUNT3 based on HUNT4 70+ (Model 1) and donation of blood samples in HUNT4 70+ (Model 2)**

| Coefficients | | Model 1 Estimation | Model 1 Std. Err. | Model 1 P-value | Model 2 Estimation | Model 2 Std. Err. | Model 2 P-value |
|---|---|---|---|---|---|---|---|
| Intercept | | 2.85 | 0.10 | <.001 | -1.19 | 0.07 | <.001 |
| Diagnosis | CU | Ref. | | | | | |
| | MCI | -0.26 | 0.09 | 0.004 | 1.83 | 0.05 | <.001 |
| | Dementia | -1.14 | 0.10 | <.001 | 4.74 | 0.18 | <.001 |
| | Other | -0.73 | 0.21 | 0.000 | -14.30 | 122.39 | 0.907 |
| Sex | Female | Ref. | | | | | |
| | Male | 0.19 | 0.07 | 0.008 | 0.03 | 0.05 | 0.569 |
| Age group | 70 - 74 | Ref. | | | | | |
| | 75 - 79 | -0.07 | 0.10 | 0.519 | -0.04 | 0.06 | 0.510 |
| | 80 - 84 | -0.53 | 0.10 | <.001 | 0.03 | 0.08 | 0.710 |
| | 85 - 89 | -0.87 | 0.11 | <.001 | -0.19 | 0.11 | 0.074 |
| | 90 + | -1.08 | 0.12 | <.001 | 0.06 | 0.15 | 0.702 |
| Education | Primary | Ref. | | | | | |
| | Secondary | 0.04 | 0.08 | 0.618 | -0.13 | 0.06 | 0.039 |
| | Tertiary | 0.28 | 0.11 | 0.010 | -0.08 | 0.07 | 0.295 |
| | Missing | -0.16 | 0.27 | 0.544 | 0.11 | 0.59 | 0.850 |
| APOE ε4 allele | ε3ε3/ε3ε2/ε2ε2 | Ref. | | | | | |
| | ε2ε4/ε3ε4 | -0.18 | 0.08 | 0.023 | 0.04 | 0.06 | 0.499 |
| | ε4ε4 | -0.31 | 0.20 | 0.122 | 0.03 | 0.17 | 0.859 |
| | Missing | -1.90 | 0.21 | <.001 | 0.27 | 0.36 | 0.459 |

CU = cognitively unimpaired; MCI = mild cognitive impairment; Std. Err. = standard error; Yr=years; *APOE* =apolipoprotein E. Model 1 estimates the probability of being selected into the HUNT3 nested study based on cognitive status at HUNT4 70+; Model 2 estimates the probability of donating a blood sample at HUNT4 70+. Estimates are presented on the logit scale; appropriate transformations were applied to obtain final probabilities for weight calculation.

# Reporting Summary

## Statistics

For all statistical analyses, confirm that the following items are present in the figure legend, table legend, main text, or Methods section.

| n/a | Confirmed | |
|---|---|---|
| ☐ | ☒ | The exact sample size (*n*) for each experimental group/condition, given as a discrete number and unit of measurement |
| ☐ | ☒ | A statement on whether measurements were taken from distinct samples or whether the same sample was measured repeatedly |
| ☐ | ☒ | The statistical test(s) used AND whether they are one- or two-sided<br>*Only common tests should be described solely by name; describe more complex techniques in the Methods section.* |
| ☐ | ☒ | A description of all covariates tested |
| ☐ | ☒ | A description of any assumptions or corrections, such as tests of normality and adjustment for multiple comparisons |
| ☐ | ☒ | A full description of the statistical parameters including central tendency (e.g. means) or other basic estimates (e.g. regression coefficient) AND variation (e.g. standard deviation) or associated estimates of uncertainty (e.g. confidence intervals) |
| ☐ | ☒ | For null hypothesis testing, the test statistic (e.g. *F*, *t*, *r*) with confidence intervals, effect sizes, degrees of freedom and *P* value noted<br>*Give P values as exact values whenever suitable.* |
| ☒ | ☐ | For Bayesian analysis, information on the choice of priors and Markov chain Monte Carlo settings |
| ☒ | ☐ | For hierarchical and complex designs, identification of the appropriate level for tests and full reporting of outcomes |
| ☒ | ☐ | Estimates of effect sizes (e.g. Cohen's *d*, Pearson's *r*), indicating how they were calculated |

*Our web collection on statistics for biologists contains articles on many of the points above.*

## Software and code

Policy information about availability of computer code

| Data collection | No software was used for data collection as part of this specific study. |
|---|---|
| Data analysis | R version 4.5.0; R Core Team, packages haven, survey, tidyverse, ggplot2, ggpubr, ggbeeswarm, patchwork, segmented. All codes for data processing and analysis are publicly available at https://github.com/dalejo643/Nature-Aarsland2025.git |

For manuscripts utilizing custom algorithms or software that are central to the research but not yet described in published literature, software must be made available to editors and reviewers. We strongly encourage code deposition in a community repository (e.g. GitHub). See the Nature Portfolio guidelines for submitting code & software for further information.

## Data

Policy information about availability of data

All manuscripts must include a data availability statement. This statement should provide the following information, where applicable:
- Accession codes, unique identifiers, or web links for publicly available datasets
- A description of any restrictions on data availability
- For clinical datasets or third party data, please ensure that the statement adheres to our policy

To protect participants' privacy, HUNT Research Centre aims to limit storage of data outside HUNT databank and cannot deposit data in open repositories. HUNT databank has precise information on all data exported to different projects and can reproduce them on request. There are no restrictions regarding data export given approval of applications to HUNT Research Centre. Researchers can apply for data at https://www.ntnu.edu/hunt/research.

# Research involving human participants, their data, or biological material

Policy information about studies with <u>human participants or human data</u>. See also policy information about <u>sex, gender (identity/presentation), and sexual orientation</u> and <u>race, ethnicity and racism</u>.

| | |
|---|---|
| Reporting on sex and gender | Sex was defined as either female or male according to the status registered in the Norwegian National Registry at the time of invitation. No information on gender was collected. |
| Reporting on race, ethnicity, or other socially relevant groupings | The HUNT study does not collect data on ethnicity, but the population in this region in 2017 included less than 5% immigrants or Norwegian-born to immigrant parents from Africa, Asia, Middle- or South America, and thus the findings are relevant for a mainly Caucasian Norwegian population. Education was self-reported. |
| Population characteristics | All participants lived in the North Trøndelag region in Norway and were 58 years or older. |
| Recruitment | Participants were recruited from the North Trøndelag region in Norway. All inhabitants in the defined geographical region were eligible for participation in the HUNT study. For this study, participants aged 58-69.9 years were recruited from HUNT3 and participants aged 70+ years were recruited from HUNT4. |
| Ethics oversight | This study was approved by the Regional Committee for Medical and Health Research Ethics in Norway (REC Southeast C 565876) as well as according to the General Data Protection Regulation by the Norwegian Agency for Shared Services in Education and Research (SIKT 585403). Participation in HUNT required informed consent, which was provided after receiving oral and written information about the health survey. In participants with reduced capacity to consent, their next of kin gave consent. |

Note that full information on the approval of the study protocol must also be provided in the manuscript.

# Field-specific reporting

Please select the one below that is the best fit for your research. If you are not sure, read the appropriate sections before making your selection.

☒ Life sciences ☐ Behavioural & social sciences ☐ Ecological, evolutionary & environmental sciences

For a reference copy of the document with all sections, see nature.com/documents/nr-reporting-summary-flat.pdf

# Life sciences study design

All studies must disclose on these points even when the disclosure is negative.

| | |
|---|---|
| Sample size | Biomarker status from all participants in HUNT4 70+ with available plasma were analysed (n=8,949 ).<br>Plasma biomarker analysis from participants in HUNT3 was restricted to n=2,537 because of funding reasons. |
| Data exclusions | No data was excluded. |
| Replication | All laboratory analyses were subject to standard Quality Control (QC) procedures. Information is included in the Methods section regarding QC of the SIMOA assays specifically. Replication of the results was not feasable because of the high number of samples. |
| Randomization | Not applicable to this type of study as it was a observational study and no interventions were assigned. |
| Blinding | Laboratory personnel performing plasma pTau217 analysis were blinded to all demographic, clinical and cognitive information on participants. |

# Behavioural & social sciences study design

All studies must disclose on these points even when the disclosure is negative.

| | |
|---|---|
| Study description | |
| Research sample | |
| Sampling strategy | |
| Data collection | |
| Timing | |
| Data exclusions | |

| Non-participation | |
|---|---|
| Randomization | |

# Ecological, evolutionary & environmental sciences study design

All studies must disclose on these points even when the disclosure is negative.

| Study description | |
|---|---|
| Research sample | |
| Sampling strategy | |
| Data collection | |
| Timing and spatial scale | |
| Data exclusions | |
| Reproducibility | |
| Randomization | |
| Blinding | |

Did the study involve field work?  ☐ Yes  ☐ No

## Field work, collection and transport

| Field conditions | |
|---|---|
| Location | |
| Access & import/export | |
| Disturbance | |

# Reporting for specific materials, systems and methods

We require information from authors about some types of materials, experimental systems and methods used in many studies. Here, indicate whether each material, system or method listed is relevant to your study. If you are not sure if a list item applies to your research, read the appropriate section before selecting a response.

### Materials & experimental systems

| n/a | Involved in the study |
|---|---|
| ☐ | ☒ Antibodies |
| ☒ | ☐ Eukaryotic cell lines |
| ☒ | ☐ Palaeontology and archaeology |
| ☒ | ☐ Animals and other organisms |
| ☐ | ☒ Clinical data |
| ☒ | ☐ Dual use research of concern |
| ☒ | ☐ Plants |

### Methods

| n/a | Involved in the study |
|---|---|
| ☒ | ☐ ChIP-seq |
| ☒ | ☐ Flow cytometry |
| ☒ | ☐ MRI-based neuroimaging |

## Antibodies

| Antibodies used | ALZpath p-Tau 217 Advantage PLUS, Quanterix |
|---|---|
| Validation | The study used a two cut-off approach to categorize individuals as Alzheimer's disease neuropathology negative, intermediate, or |

| | |
|---|---|
| Validation | positive. Cut-offs were validated by Ashton et al (doi:10.1001/jamaneurol.2023.5319). The lower cut-off was 0.40 pg/mL (95% sensitivity) and the upper cut-off was ≥ 0.63 pg/mL (95% specificity). |

## Eukaryotic cell lines

Policy information about cell lines and Sex and Gender in Research

| | |
|---|---|
| Cell line source(s) | |
| Authentication | |
| Mycoplasma contamination | |
| Commonly misidentified lines (See ICLAC register) | |

## Palaeontology and Archaeology

| | |
|---|---|
| Specimen provenance | |
| Specimen deposition | |
| Dating methods | |

☐ Tick this box to confirm that the raw and calibrated dates are available in the paper or in Supplementary Information.

| | |
|---|---|
| Ethics oversight | |

Note that full information on the approval of the study protocol must also be provided in the manuscript.

## Animals and other research organisms

Policy information about studies involving animals; ARRIVE guidelines recommended for reporting animal research, and Sex and Gender in Research

| | |
|---|---|
| Laboratory animals | |
| Wild animals | |
| Reporting on sex | |
| Field-collected samples | |
| Ethics oversight | |

Note that full information on the approval of the study protocol must also be provided in the manuscript.

## Clinical data

Policy information about clinical studies
All manuscripts should comply with the ICMJE guidelines for publication of clinical research and a completed CONSORT checklist must be included with all submissions.

| | |
|---|---|
| Clinical trial registration | NCT06719453 |
| Study protocol | Information on HUNT can be found on https://hunt-db.medisin.ntnu.no/hunt-db/ |
| Data collection | The Trøndelag Health Study (HUNT) is a population-based health study conducted in the Trøndelag region of Central Norway and has so far spanned four waves (HUNT1-4). Data used in this study was collected as part of HUNT wave 3 (10/2006-6/2008) and wave 4 (9/2017-2/2019). Data collected included demographic information, clinical information and blood samples. Additionally, in HUNT4, all residents aged 70 years or older were invited to the substudy HUNT4 70+ for a standardized cognitive assessment and diagnosis. Trained health personnel assessed the participants' cognitive, neuropsychiatric, and functional status using standardized clinical scales at a field station, at homes, or in nursing homes. Clinical and research experts made diagnoses on cognition by clinical consensus method. This study collected data on plasma pTau217, with biomarker analysis conducted 1-8/2024, and otherwise used previously collected data. |
| Outcomes | The outcome was the plasma pTau217 concentration. Plasma pTau217 concentrations were measured using the ALZpath p-Tau 217 Advantage PLUS kit and the Simoa HD-X instrument (both Quanterix - Billerica, MA, USA). We used a two cut-off approach to categorize individuals as Alzheimer's Disease Neuropathological Changes (ADNC) negative, intermediate, or positive, using a lower |

cut-off of 0.40 pg/mL (95% sensitivity) and an upper cut-off of ≥ 0.63 pg/mL (95% specificity). We used the upper cut-off to determine with high specificity those individuals with the presence of ADNC (those above the upper cut-off), and the lower cut-off to identify those with a high likelihood of not having ADNC (those below the lower cut-off). The concentrations between the two cut-offs were not able to inform on the presence of ADNC.

# Dual use research of concern

Policy information about dual use research of concern

## Hazards

Could the accidental, deliberate or reckless misuse of agents or technologies generated in the work, or the application of information presented in the manuscript, pose a threat to:

No | Yes
☐ ☐ Public health
☐ ☐ National security
☐ ☐ Crops and/or livestock
☐ ☐ Ecosystems
☐ ☐ Any other significant area

## Experiments of concern

Does the work involve any of these experiments of concern:

No | Yes
☐ ☐ Demonstrate how to render a vaccine ineffective
☐ ☐ Confer resistance to therapeutically useful antibiotics or antiviral agents
☐ ☐ Enhance the virulence of a pathogen or render a nonpathogen virulent
☐ ☐ Increase transmissibility of a pathogen
☐ ☐ Alter the host range of a pathogen
☐ ☐ Enable evasion of diagnostic/detection modalities
☐ ☐ Enable the weaponization of a biological agent or toxin
☐ ☐ Any other potentially harmful combination of experiments and agents

# Plants

Seed stocks

Novel plant genotypes

Authentication

# ChIP-seq

## Data deposition

☐ Confirm that both raw and final processed data have been deposited in a public database such as GEO.

☐ Confirm that you have deposited or provided access to graph files (e.g. BED files) for the called peaks.

Data access links
*May remain private before publication.*

Files in database submission

Genome browser session
(e.g. UCSC)

## Methodology

Replicates

Sequencing depth

Antibodies

Peak calling parameters

Data quality

Software

# Flow Cytometry

## Plots

Confirm that:

☐ The axis labels state the marker and fluorochrome used (e.g. CD4-FITC).

☐ The axis scales are clearly visible. Include numbers along axes only for bottom left plot of group (a 'group' is an analysis of identical markers).

☐ All plots are contour plots with outliers or pseudocolor plots.

☐ A numerical value for number of cells or percentage (with statistics) is provided.

## Methodology

Sample preparation

Instrument

Software

Cell population abundance

Gating strategy

☐ Tick this box to confirm that a figure exemplifying the gating strategy is provided in the Supplementary Information.

# Magnetic resonance imaging

## Experimental design

Design type

Design specifications

Behavioral performance measures

## Acquisition

Imaging type(s)

Field strength

Sequence & imaging parameters

Area of acquisition

Diffusion MRI          ☐ Used          ☐ Not used

## Preprocessing

Preprocessing software

Normalization

Normalization template

Noise and artifact removal

Volume censoring

## Statistical modeling & inference

Model type and settings

Effect(s) tested

Specify type of analysis:  ☐ Whole brain  ☐ ROI-based  ☐ Both

Statistic type for inference

(See Eklund et al. 2016)

Correction

## Models & analysis

| n/a | Involved in the study |
| --- | --- |
| ☐ | ☐ Functional and/or effective connectivity |
| ☐ | ☐ Graph analysis |
| ☐ | ☐ Multivariate modeling or predictive analysis |

Functional and/or effective connectivity

Graph analysis

Multivariate modeling and predictive analysis

