## [Peer Review file · Nature]

Prevalence of Alzheimer's Disease Pathology in the Community

Corresponding Author: Ms Anita Sunde

Reviewer #2 was unavailable for re-review. The authors' responses to Reviewer #2's comments were assessed by one of the other reviewers.

Version 0:

Reviewer comments:

Referee #1

(Remarks to the Author)

Key findings. This is a large, population-based investigation of the prevalence of Alzheimer's disease-related pathology, using the P-tau217 biomarker from blood. This is the largest study of its kind focused on a representative cohort of individuals over age 57 years from Norway. Not unsurprisingly the frequency of elevated P-tau levels is increased with advancing age, with lower educational achievement, with possession of one or more APOE-e4 alleles and with comorbidities, specifically chronic renal and heart disease. How comorbidities affect the results is not clear in the manuscript.

Validity. The authors used the "two cut-off approach" published by the Global CEO Initiative on Alzheimer's disease (Schindler et al. Nat Rev Neurol 2024). Establishing optimal cut-points has been the norm for most studies of this kind. They need not be elaborate, but it would have been a bit more convincing had they demonstrated that elevation of P-tau217 past the cut point was associated with either autopsy confirmed Alzheimer's disease or cerebral amyloidosis on PET scan. Ultimately P-tau cut points vary based on the specific assay and platform. The paper would be strengthened by using cut-points specific to the population. Among those with dementia only 57.7% had elevated P-tau217.

Originality and significance. The main findings in the study, that age, APOE genotype and comorbidities affect the level of P-tau217 are not new. The importance of the study is the population level data.

Data and methodology. I have already commented on the need to establish specific cut points for the cohort. It is not clear how the diagnoses of dementia, mild cognitive impairment and no dementia were established. That information could go into the supplemental data. The authors mention criteria for diagnoses but do not specify which test platform was used.

Statistics. The statistical approach was appropriate. In the supplemental data they show the effect sizes for the various comorbidities. It would be useful to also present the frequency of ADNC, dementia and mild cognitive impairment stratified by the most frequent comorbid disorders: renal and heart disease.

Conclusions. I found no faults with the conclusions and data interpretation. However, they could have conducted more analyses on difference by sex, APOE genotype and comorbidities. They have the sample size need for such analyses.

Suggested improvements.

1. Validate the P-tau217 in your population.
2. In depth analyses of the effects of comorbidities on the diagnoses and the presence of ADNC.

References. Most of the important papers were included.

Clarity and context. The abstract, introduction and conclusions appropriate for the study, but require a bit more detail in areas specified above.

Referee #2

(Remarks to the Author)

This observational study by Aarsland et al. provides important estimates of Alzheimer disease neuropathological changes (ADNC) as assessed with plasma pTau217 in over 11,000 participants aged 58 years and older from the population-based

Norwegian Trøndelag Health study (HUNT study). ADNC was determined using a two-threshold approach based on plasma pTau217 cutoffs (categorized as ADNC negative (<0.40 pg/mL), intermediate, or positive (≥ 0.63 pg/mL) based upon a previous validation study in observational cohorts. The study provides valuable data on the prevalence of ADNC, as well as preclinical AD, prodromal AD, and AD dementia, which were further compared by age groups, sex, and APOE genotype. In addition, the authors proposed and examined an algorithm to identify potential candidates for disease-modifying therapies (DMTs) and found that 10% of the 70+ population met eligibility criteria. This is a novel and timely contribution to the field, offering the largest community-based estimate of biomarker-defined AD burden, with potential clinical and public health implications. The authors are to be applauded on their efforts, however, there are several aspects of the study/manuscript that need attention and several methodological limitations need to be addressed. In particular the sampling strategy, low participation (46%) in study from HUNT 4 and biases that that introduces, oversampling of MCI and dementia from HUNT 3, make the interpretation of the results extremely challenging.

1. While pTau 217 has emerged as the biomarker of choice of AD, it is surprising that amyloid beta 42 was not also assessed. This especially relevant given the use of the ratio of pTau 217 to A 42. Why was A not measured and how does this limit this investigation?
2. Another major issue is the definition of ADNC solely based on plasma pTau217 without within-sample validation using PET or CSF biomarkers. The pTau217 cutoffs used in the current study were derived from previous validation in subsamples from three observational studies enriched with cognitively impaired individuals. Given differences in sample composition, especially regarding the prevalence of ADNC and cognitive impairment, the positive predictive value may differ when applied to a community-based sample such as HUNT. The authors should expand the discussion on how this may have affected their prevalence estimates.
3. The clinical evaluation in the HUNT study was limited to those aged 70 plus. For cost and efficiency, this makes sense but it is therefore impossible to know the clinical status of those younger who may have prodromal AD.
4. There is too much focus on the explicit goal to identify the % in need for DMT medications that have not been approved in many countries and are not recommended, and may never be, for asymptomatic adults. This is especially concerning given almost certain selection bias.
5. It is critical to show the positive predictive value and negative predictive value by age strata. This age range is a big strength of the study but is pretty much ignored. I realize this can only be done 70 plus but be good to see.
6. The Figure 3 y axis should be 100%.
7. There does not seem to be any subtype of dementia or MCI so it is hard to know if clinical AD or other clinical subtypes. There may be important vascular components to dementia that also could be associated with ADNC and the definition here of AD is limited to pTau217 with no ability to look at mixed dementia types or even association of pTau217 to other pathologies.
8. Although the two-cutoff approach is recommended, the large intermediate group (pTau217 0.40–0.63 pg/mL) in up to 28% of the cohort highlights the challenge of classifying ADNC in the general population. The authors need to address the handling and interpretation of this intermediate group; otherwise, the advantage of using the two-cutoff approach is unclear.
9. Another very big concern is the selection bias of who agreed to clinical cognitive evaluation/blood assay from HUNT 4 and how this bias could explain higher prevalence of clinical syndromes. The 46% participation is actually quite poor despite the large absolute numbers and the authors view that this is a success. The authors do use a weighting to try to get at this issue but it is a pretty bare-boned approach. The weighting should be expanded include other variables that differ among those who agreed vs did not including variables such as APOE e4, age, sex, medical history etc. One also could conduct an inverse probability weighting on those.
10. Greater explanation is needed on the 20% selection of younger participants. The fact that they oversampled of MCI and dementia (from HUNT3 but ascertained in HUNT4) makes the Fig 3 prevalence of clinical status impossible to interpret. I think this needs to be changed. Also Figure 1 is misleading given the oversampling. The sampling limitations, unfortunately, are really critical to the interpretations here. If a senior biostatistician has not been involved, that would be helpful.

Minor issues:

1. It appears that sample freezing occurred the next day and the needle-to-freezer time exceeded 2 hours. Can the authors clarify how this may affect the stability and accuracy of the assays?
2. A brief discussion of assay variability (e.g., inter-assay CVs) would be informative.
3. It is unclear why serum creatinine was used instead of eGFR, which is generally considered more accurate and clinically meaningful for assessing kidney function.

Referee #3

(Remarks to the Author)

A. Summary of Key results

In a cross-sectional study nested in a large Norwegian population-based study, the authors examined the proportions of persons with elevated plasma-tau (ADpTAU), based on standard cut-offs. These proportions increased with age and with worse cognitive performance. As would be anticipated, age is the key variable to predict elevated plasma-tau.

B. Originality and significance

This work represents a careful study in a population-based sample. Few such large samples have the necessary blood specimens preserved, as well as cognitive classification. This work is an important contribution to the study of this highly relevant biomarker.

C. Data and methodology

Generally, the statistics are appropriate, except for inconsistent control for age. It is clear (as would be expected) that age-at-blood draw is a key driver of the distribution of phosphorylated tau.; controlling for age in comparisons of sex or of diagnosis groups would improve the rigor of these analyses.

Please state even more clearly that this work can be thought of as an analysis of data at one visit (the visit) of the blood draw and study data for visits selected from the longitudinal study. For the younger participants in HUNT3, subsequent diagnostic history was considered when deciding who to approach for inclusion.

D. Conclusions:

D1. It is important to point out that plasma measures cannot identify the spatial distribution of phosphorylated-tau. The spatial distribution of tau is essential for distinguishing tauopathies that are not Alzheimer's Disease (AD), including FrontoTemporal Lobar Degeneration (FTLD), Chronic Traumatic Encephelopathy (CTE), and Primary Age-Related Tauopathy (PART), from AD. Data for the prevalence of these other tauopathies is even sparser than for AD, but these neurodegenerative patterns that are distinct from AD should be mentioned.

One early publication describing PART:

Crary, John F.; Trojanowski, John Q.; Schneider, Julie A.; Abisambra, Jose F.; Abner, Erin L.; Alafuzoff, Irina; Arnold, Steven E.; Attems, Johannes; Beach, Thomas G. (2014-12-01). "Primary age-related tauopathy (PART): a common pathology associated with human aging". *Acta Neuropathologica*. 128 (6): 755–766. doi:10.1007/s00401-014-1349-0. ISSN 0001-6322. PMC 4257842. PMID 25348064.

D2. The scientific rigor of this work in the study of biomarkers for AD would be substantially improved if the terminology were more precise throughout. The plasma-tau biomarkers do indeed show promising associations (AUC around 0.8) with measures from CSF or from neuropathology, but these findings from different biospecimens are not interchangeable. To their credit, the authors avoid the confusion between Alzheimer's Disease (AD) (defined as a biologic process) and AD-Dementia (the clinical recognition of dementia of the Alzheimer's type). However, describing plasma biomarkers as ADNC (Alzheimer's Disease Neuropathologic Change) is a step toward imprecision.

E. Suggested improvements:

E1: I understand this study to be a nested cross-sectional study of participants in one of two waves (HUNT3, HUNT4) of a large Norwegian cohort study in which exactly one blood draw was analysed. Since 2,391 individuals participated in both HUNT3 and HUNT4; please clarify whether one blood biospecimen or two biospecimens were studied. I assume one biospecimen per participant, as I do not see any methodologic allowance for repeated ADp-TAU values

E2: Tight control for age (at least in the 5-year intervals, but ideally in 1-year intervals) is essential when comparing sex or diagnosis groups or medical comorbidities.

E3: Figure 2: Age is clear confounding variable in the comparison of the diagnosis groups, as the diagnosis distributions shift with age. Consider repeating Figure 1 separately for each of the 3 diagnosis groups; such figures might well improve the clarity of the comparison. would be an improvement.

E4. As described above (point D), I recommend removing "ADNC" from most of the manuscript, with the possible exception of discussion.

E5. Minor point: A brief explanation in the supplementary material of the reason a very small number of persons were not classified into a cognitive status. Does this refer to the inability to get a consensus diagnosis? Or patterns of dementia not consistent with AD? Or incomplete testing?

G. Clarity and context: lucidity of abstract/summary, appropriateness of abstract, intro conclusions.

G1. See earlier points on the distinction between biomarker positivity and ADNC

G2. The vascular risk factors could be discussed in the context of vascular disease, especially with the recognition that vascular disease can both cause dementia and increase the odds that AD pathology manifests as AD dementia.

G3. The authors recognize the potential differences of different ethnic and racial groups, and recognize that these non-Caucasian groups are not well-represented in Norway. One subtle change would be to refer to "populations" rather than "population" throughout the introduction and conclusion. A rigorous evaluation of one population is very valuable. The Norwegian population may differ even from other Caucasian populations in many ways, including in the presence of risk factors for vascular disease. Since mixed pathologies are important in the manifestation of dementia in older persons, this clear picture of Norway may not automatically translate to other countries.

Version 1:

Reviewer comments:

Referee #1

(Remarks to the Author)
2025-02-04429A

A. Summary of the key results. This study examined the prevalence of neuropathological changes suggestive of Alzheimer's disease by using plasma P-tau 217. The strength of the paper is the measurement of this biomarker in over 11,000 individuals from a Norwegian population-based cohort. Remarkably the estimated prevalence conforms to what was already known (Jama neurology 2003;60;(8):1119-1122). The percentage of individuals with Alzheimer's increases from approximately 8% around the age of 62 over 60% by age 90. The contribution of this study is that we now have a biomarker that can confirm what was previously established using clinical methods for diagnosis.

B. Originality and significance: The novel approach of using plasma P-tau 217 in the community is the strength of the manuscript and allows for a biological basis of understanding the impact Alzheimer's will have on the aging population. The authors do make the point that based on their findings a significant number of individuals would be eligible for monoclonal antibody therapy. However, this comes with risks that would have to be weighed in conjunction with any substantial benefit from this type of intervention.

C. Data & methodology: validity of approach, quality of data, quality of presentation. The paper is well written with excellent methodology and careful attention to details. It is also presented in a clear fashion. One issue that concerns me is the observation by others that plasma P-tau217 does increase with age in the absence of dementia or any neurological disease. The authors might want to comment on that or if they have data show the degree of increase in comparison to those who develop Alzheimer's disease later in life.

On line 219 the authors refer to further examinations should values of fall in a certain range among younger individuals. They do not specify what type of examinations to be conducted. However, on line 219 they do mention lumbar puncture for cerebrospinal fluid analysis and amyloid PET scan. It would be helpful if they clarified that this was the type of examination they are referring to.

D. Appropriate use of statistics and treatment of uncertainties. These statistics were appropriate however I do have questions about the different cut points that were used by age group. Was this chosen to adjust for P-tau changes with age or with some other decision made a priori.

E. Conclusions: robustness, validity, reliability. I believe that the conclusions are supported by the work presented and I have no further comments or suggestions other than those listed above.

Richard Mayeux

Referee #3

(Remarks to the Author)

I confine my comments to changes.

A. Summary: The key results remain.

B. Originality and significance: This work remains an important contribution to the study of the p-tau plasma biomarkers.

C. The data and methodology are appropriate, especially since the scientific rigor has been improved by emphasizing that the assessment of ADNC is based on the plasma marker. The emphasis is not in the abstract or title, but is clear in the methods and is included in the caption for each figure.

D. The methodology for control of age has been improved, so that all statistical methods are appropriate. Error bars and other details in the figures are described completely and clearly. Uncertainties are described clearly and appropriately.

E. The conclusions are clear and sound.

F. The authors have made substantial improvements in response to the reviews. I commend the authors for the clarity and successful accomplishment of these changes I have no other improvements to suggest.

We are grateful to the reviewers for their thoughtful and knowledgeable feedback, which has significantly strengthened our manuscript. In response, we have provided a detailed point-by-point reply. The reviewers' comments are shown in black, while the authors' response is shown in blue. Place of change in the track-change document is indicated in green writing.

Reply to reviewer 1

1. How comorbidities affect the results is not clear in the manuscript. Recommend in depth analyses of the effects of comorbidities on the diagnoses and the presence of ADNC.

We thank the reviewer for the suggestion to conduct a more in-depth analysis of the relationship between comorbidities and ADNC (plasma pTau217 \geq 0.63 pg/mL). We have performed a multivariable logistic regression analysis to evaluate the association between each self-reported comorbidity and the prevalence of ADNC, adjusting for age, sex, APOE ϵ 4 allele count, creatinine and education level. The analysis was performed by using age (5-year intervals and 1-year intervals). After re-analysis, the association between ADNC and comorbidities was no longer significant.

Place of change in the track-change document:

Manuscript page 7, 3rd paragraph

Manuscript page 9, 2nd paragraph

Manuscript page 20, last paragraph

Supplementary Table 7

2. Establishing optimal cut-points has been the norm for most studies of this kind. They need not be elaborate, but it would have been a bit more convincing had they demonstrated that elevation of P-tau217 past the cut point was associated with either autopsy confirmed Alzheimer's disease or cerebral amyloidosis on PET scan. Ultimately P-tau cut points vary based on the specific assay and platform. The paper would be strengthened by using cut-points specific to the population.

We appreciate the reviewer's concern. These cut-offs have been widely validated against amyloid PET, in the original (Ashton et al., JAMA Neurology, 2024) and in following papers which span various population groups (<https://academic.oup.com/brain/article/148/2/408/7921772>, <https://pmc.ncbi.nlm.nih.gov/articles/PMC11503049/>, <https://www.sciencedirect.com/science/article/pii/S2352396424004493>), with longitudinal stability in the measurements ascertained using the same internal control samples. Moreover, the ALZpath test has been implemented as an LDT (Laboratory Developed Test) with its performance characteristics determined by Neurocode, USA Inc. at Northwest Pathology, P.S., dba Avero Diagnostics, adopting similar cut-offs as

were used in our study (>0.37 and <0.63). Lastly, it should be noted that the FDA-approved blood-based cut-offs for AD are validated within each population.

Place of change in the track-change document:

Manuscript page 4, first paragraph, new reference PMID32722745

3. It is not clear how the diagnoses of dementia, mild cognitive impairment and no dementia were established. That information could go into the supplemental data. The authors mention criteria for diagnoses but do not specify which test platform was used.

We appreciate the reviewer pointing this out and have now specified this more in depth.

Place of change in the track-change document:

Manuscript, page 21, first paragraph

New Supplementary Material_Assessment of cognition, physical performance, anxiety, depression, neuropsychiatric symptoms, activities of daily living and cognitive diagnosis

Reply to reviewer 2

1. While pTau 217 has emerged as the biomarker of choice of AD, it is surprising that amyloid beta 42 was not also assessed. This especially relevant given the use of the ratio of pTau 217 to A β 42. Why was A β not measured and how does this limit this investigation?

We thank the reviewer for raising this important point, which is certainly relevant in the context of the recently FDA-approved plasma p-tau₂₁₇/A β ₄₂ test by Fujirebio. However, several scientific, practical, and logistical considerations informed our choice to use plasma p-tau₂₁₇ alone in this study.

First, the timing and logistics of the HUNT study are critical to understand. The plasma analyses commenced in January 2024 and required eight months to complete. At that time, the only commercially available and independently validated assay was the ALZpath p-tau₂₁₇ test. A commercial test was essential to ensure large batch lots could be produced specifically for this study. The Lumipulse p-tau₂₁₇ standalone assay was not yet available, and a validated plasma A β ₄₂ assay was lacking. Therefore, the inclusion of A β ₄₂ was simply not feasible.

Second, while the p-tau₂₁₇/A β ₄₂ ratio can reduce intermediate test results and may slightly increase diagnostic accuracy in well-controlled research cohorts (Palmqvist et al., Nat Med 2025; Wang et al., Alzheimers Dement 2025; Lehmann et al., eBioMed 2025), these findings do not directly translate to large population-based studies like HUNT. Plasma A β ₄₂ is highly sensitive to pre-analytical variables—including storage

duration, centrifugation, and temperature—which often deviate from Alzheimer's Association guidelines in population cohorts (Sunde et al., *Alzheimers Dement* 2023).

Moreover, plasma A β 42 originates from both central and peripheral sources (PMID: 31371569), causing a poor correlation with CSF A β 42. Peripheral contributions dilute the A β 42 signal, leading to minimal (~10%) differences in plasma A β 42/A β 40 ratios between amyloid-positive and negative individuals (PMID: 34542571), making the marker prone to misclassification under real-world variability (PMID: 36150024; PMID: 35130933). It has been demonstrated that even small variations in the plasma A β measurements lead to a reduction in accuracy from 85% to <70% (PMID: 35130933). Similar results were recently published from the Mayo Clinic using the FDA-approved assays (PMID: 40308118). Additionally, plasma A β levels are significantly influenced by common medications, such as neprilysin inhibitors, which increase peripheral A β independent of brain amyloid pathology (PMID: 38109077).

All these factors have made the International Federation of Clinical Chemistry and Laboratory Medicine (IFCC) Biomarkers for Neurodegenerative Diseases (BND) Working Group to focus their standardization work on p-tau217 alone, and not on a ratio with A β 42, since the latter is not considered clinically robust.

Place of change in the track-change document: Manuscript page 8, 3rd paragraph

2. Another major issue is the definition of ADNC solely based on plasma pTau217 without within-sample validation using PET or CSF biomarkers. The pTau217 cutoffs used in the current study were derived from previous validation in subsamples from three observational studies enriched with cognitively impaired individuals. Given differences in sample composition, especially regarding the prevalence of ADNC and cognitive impairment, the positive predictive value may differ when applied to a community-based sample such as HUNT. The authors should expand the discussion on how this may have affected their prevalence estimates.

We agree and have expanded the Discussion to emphasize that our ADNC prevalence estimates rely on plasma p-tau217 cutoffs previously validated in enriched cohorts and are not anchored to PET/CSF within HUNT. To gauge how differences in disease prevalence might affect predictive performance in a community-based setting, we now include an exploratory PPV/NPV sensitivity analysis.

Briefly, we combined externally reported sensitivity/specificity ranges for p-tau217 (Ashton et al.) with age-stratum weighted prevalence of p-tau217 positivity in HUNT4 (≥ 70 years). Across age strata, PPV rose with higher prevalence (e.g., 59.9% at 70–74 to 92.6% at 90+), while NPV declined (97.9% to 84.9%). Importantly, this analysis is not a substitute for internal validation and has key limitations that we now state clearly: (i) commutability of sensitivity/specificity from external cohorts is assumed; (ii)

circularity is possible because prevalence is estimated from the same biomarker; (iii) we do not model age-specific shifts in sensitivity/specificity or spectrum effects.

Place of change in the track-change document:

Manuscript page 6, 2nd paragraph

Manuscript page 9, 3rd paragraph

Manuscript page 24, section on Predictive-value sensitivity analysis

New Supplementary Table 8

3. The clinical evaluation in the HUNT study was limited to those aged 70 plus. For cost and efficiency, this makes sense but it is therefore impossible to know the clinical status of those younger who may have prodromal AD.

We agree with the reviewer that this is a limitation. Categorization of those with elevated plasma pTau217 into preclinical AD, prodromal AD and AD dementia was only conducted in those with available cognitive assessment, namely those aged 70 plus (see Methods section). We have emphasized this better in the Discussion section.

Place of change in the track-change document: Manuscript page 6, 4th paragraph

4. There is too much focus on the explicit goal to identify the % in need for DMT medications that have not been approved in many countries and are not recommended, and may never be, for asymptomatic adults. This is especially concerning given almost certain selection bias.

In our analysis on DMT eligibility, we only included those participants with a clinical diagnosis of mild cognitive impairment or mild dementia and not those with preclinical (asymptomatic) AD. We think this is still a relevant topic, also since lecanemab was approved by the EMA for patients with prodromal AD and mild Alzheimer's disease dementia after submission. We have however reduced the text about this in the Main text and Conclusion.

Place of change in the track-change document:

Manuscript page 7, first line

Manuscript page 10, first paragraph

5. It is critical to show the positive predictive value and negative predictive value by age strata. This age range is a big strength of the study but is pretty much ignored. I realize this can only be done 70 plus but be good to see.

We appreciate this suggestion and have added the requested age-stratified PPV/NPV summary for participants aged ≥ 70 years (HUNT4 70+). Using the external sensitivity/specificity ranges and the weighted prevalence within each age band,

median PPV increased from 59.9% (70–74) to 92.6% (90+), while median NPV decreased from 97.9% to 84.9%; overall medians were 77.4% (PPV) and 95.4% (NPV). Because performance typically attenuates upon external validation, we also report an optimism-corrected analysis (shrinking Se/Sp 10% toward 0.5): overall medians were 71.9% (PPV) and 92.9% (NPV), with age-stratum patterns preserved. These results are presented in the main Results and detailed in Supplementary Table 8.

Place of change in the track-change document:

Manuscript page 6, 2nd paragraph

Manuscript page 9, 3rd paragraph

Manuscript page 24, section on Predictive-value sensitivity analysis

New Supplementary Table 8

6. The Figure 3 y axis should be 100%.

We thank the reviewer for pointing this out and have adjusted the scaling to 100%.

Place of change in the track-change document:

Manuscript page 18, new Figure 3

7. There does not seem to be any subtype of dementia or MCI so it is hard to know if clinical AD or other clinical subtypes. There may be important vascular components to dementia that also could be associated with ADNC and the definition here of AD is limited to pTau217 with no ability to look at mixed dementia types or even association of pTau217 to other pathologies.

This is a relevant question. The requested information is available but was not mentioned in our first draft. In HUNT4 70+, participants with dementia were further subclassified by consensus as: Alzheimer's disease dementia, vascular dementia, frontotemporal dementia, Lewy body dementia (including dementia with Lewy bodies and Parkinson's disease dementia), mixed dementia, other specified dementia, or unspecified dementia. Mild cognitive impairment was further subclassified as amnesic MCI and nonamnesic MCI (GjØra et al., 2021, J Alzheimers Dis). We have now included this information and added an analysis of estimated ADNC prevalence in the dementia and MCI subgroups. We have also commented specifically on vascular components in the discussion part.

Place of change in the track-change document:

Manuscript page 4, last paragraph, last sentence

Manuscript page 7, second-last paragraph

Manuscript page 21, first paragraph

New Supplementary Table 4. Weighted estimated proportions of plasma pTau217 categories (ADNC negative, intermediate, positive) by cognitive subgroups in people 70+ years.

New Supplementary Material_Assessment of cognition, physical performance, anxiety, depression, neuropsychiatric symptoms, activities of daily living and cognitive diagnosis

8. Although the two-cutoff approach is recommended, the large intermediate group (pTau217 0.40–0.63 pg/mL) in up to 28% of the cohort highlights the challenge of classifying ADNC in the general population. The authors need to address the handling and interpretation of this intermediate group; otherwise, the advantage of using the two-cutoff approach is unclear.

We thank the reviewer for this important point. The presence of an intermediate zone—here defined as plasma p-tau217 values between 0.40 and 0.63 pg/mL—is an expected and intrinsic feature of continuous biomarker distributions when applied to binary clinical outcomes like AD neuropathologic change (ADNC). Rather than a limitation, this zone reflects biological and clinical heterogeneity, particularly in population-based cohorts. We believe that the two-cutoff approach improves clinical utility by providing high-confidence "positive" and "negative" cases while explicitly acknowledging cases that warrant further clinical assessment or follow-up testing. In an ideal scenario, individuals in the intermediate group would undergo CSF or PET imaging to obtain a definitive diagnosis. However, recognizing that such evaluations are often not routinely available in clinical practice, we recommend a shorter follow-up interval for these individuals to allow timely reassessment and monitoring. This strategy has been outlined by the CEOi on blood biomarkers (<https://doi.org/10.1038/s41582-024-00977-5>). We have included a more elaborate statement on this in the discussion section.

Place of change in the track-change document:

Manuscript page 4, last paragraph

Manuscript page 8, last paragraph

9. Another very big concern is the selection bias of who agreed to clinical cognitive evaluation/blood assay from HUNT 4 and how this bias could explain higher prevalence of clinical syndromes. The 46% participation is actually quite poor despite the large absolute numbers and the authors view that this is a success. The authors do use a weighting to try to get at this issue but it is a pretty bare-boned approach. The weighting should be expanded include other variables that differ among those who agreed vs did not including variables such as APOE e4, age, sex, medical history etc. One also could conduct an inverse probability weighting on those.

We thank the reviewer for raising this important point. To address potential selection bias, we implemented an inverse probability weighting (IPW) approach that considers multiple stages of inclusion:

- First, we applied participation weights for HUNT4 from Skirbekk et al. (2022, <https://doi.org/10.1177/08982643221131926>), which account for age, sex, and education.
 - Second, we estimated the probability of donating blood (required for pTau217 analysis) using a logistic model that includes age, sex, APOE ϵ 4 status, cognitive diagnosis, and medical history.
 - Third, for participants in HUNT3, we additionally modelled the probability of being selected based on their HUNT4 status, using a logistic model that includes age, sex, APOE ϵ 4 status, cognitive diagnosis, and medical history.
- The final weight was computed as the inverse of the product of these probabilities. While the covariates used across models are similar, each model captures a distinct stage in the selection process. We acknowledge that this approach assumes conditional independence between stages and have now clarified this in the manuscript in the Selection Bias and Weighting subsection of the Statistical Analysis section. The added inverse probability weighting is now incorporated in the Supplementary Tables 2-6.

Place of change in the track-change document:

Manuscript page 9, 1st paragraph, last sentence

Manuscript page 20, section on Cohort Selection and Study Design

Manuscript page 22, section on Selection Bias and Weighting

Supplementary Table 9

The added inverse probability weighting is now incorporated in the Supplementary Tables 2-6, Figures 1-3 and Supplementary Figures 2 and 4.

Results are adapted in the text throughout the manuscript.

10. Greater explanation is needed on the 20% selection of younger participants. The fact that they oversampled of MCI and dementia (from HUNT3 but ascertained in HUNT4) makes the Fig 3 prevalence of clinical status impossible to interpret. I think this needs to be changed. Also Figure 1 is misleading given the oversampling. The sampling limitations, unfortunately, are really critical to the interpretations here. If a senior biostatistician has not been involved, that would be helpful.

We appreciate the reviewer's concern regarding potential selection bias. To clarify, the inverse probability weighting approach we had applied in the manuscript was implemented solely for the HUNT3 sample. This subsample was retrospectively selected based on participants' cognitive diagnoses obtained in HUNT4, and thus subject to selection bias. In contrast, the HUNT4 70+ cohort included all individuals aged 70 or older who participated in the study and provided blood samples (n = 8,949) and therefore was not subject to the same targeted selection mechanism.

As suggested, we conducted an expanded logistic regression model to estimate the probability of selection into the nested HUNT3 plasma sample group. The model

included diagnosis (cognitively unimpaired, mild cognitive impairment, dementia, or other), age, sex, APOE ϵ 4 allele count, education level, and a range of comorbidities (angina, myocardial infarction, heart failure, atrial fibrillation, stroke, diabetes, cancer, kidney disease, gout). The results confirmed that diagnosis was the only statistically significant predictor of selection. All other covariates, including APOE ϵ 4 status and age, showed no significant association with selection probability.

Furthermore, the predicted probabilities from the logistic model closely matched the simple empirical proportions used in our original weighting scheme by diagnostic group. For example, the average predicted probability of selection was 21.6% for cognitively unimpaired individuals, 63.3% for those with MCI, and 96.9% for those with dementia — virtually identical to the weights reported in (previous)

Supplementary Table 7. This supports the robustness of our original approach and suggests that additional covariates do not contribute meaningfully to explaining selection into the nested HUNT3 study.

In summary, we believe the current weighting approach appropriately addresses the selection bias present in the nested HUNT3 group, and further complexity does not substantively alter the estimates. We have added clarifying language to the methods section to make the scope and rationale of our weighting procedure more explicit.

Place of change in the track-change document:

Manuscript page 20, section on Cohort Selection and Study Design

Manuscript page 22, section on Selection Bias and Weighting

Supplementary Table 9

Minor issue 1: It appears that sample freezing occurred the next day and the needle-to-freezer time exceeded 2 hours. Can the authors clarify how this may affect the stability and accuracy of the assays?

We thank the author for addressing preanalytical sample handling conditions, which is an important consideration. In both HUNT3 and 4, preanalytical sample handling time at room temperature was maximum 120 minutes, with further storage being at the temperature of 2-8°C. In HUNT4 all, and in HUNT3 those samples collected Mondays to Thursdays, were frozen to -80 °C the consecutive day. In HUNT3, blood samples collected on Fridays were frozen to -80 °C on Mondays. It has been shown that plasma pTau217 remains stable over 72 hours with < 10% deviation from baseline, both stored at 4°C and 23°C (Figdore et al., *Alzheimers Dement*, 2025). This is in line with preanalytical stability found for other plasma pTau isoforms, who have shown no decline in pTau concentration when stored up to 24 hours at 4°C or 18°C (Sunde et al., *Alzheimers Dement*, 2023). We therefore believe that the preanalytical handling conditions have been satisfactory. These preanalytical sample handling conditions re-enforce the exclusion of A β 42 from this study.

Place of change in the track-change document:

Manuscript page 21, section on Blood sample collection and handling procedures.

Minor issue 2: A brief discussion of assay variability (e.g., inter-assay CVs) would be informative.

We have added this to the Supplementary.

Place of change in the track-change document:

New «Supplementary Material_ALZpath pTau217 analytical performance»

Minor issue 3: It is unclear why serum creatinine was used instead of eGFR, which is generally considered more accurate and clinically meaningful for assessing kidney function.

We agree with the reviewer that eGFR can give a more clinically meaningful assessment of kidney function than serum creatinine alone and thank the reviewer for pointing this out. We have now used eGFR instead of creatinine (CKD-EPI Creatinine Equation, Inker NEJM 2021, DOI: 10.1056/NEJMoa2102953), without this altering the results.

Place of change in the track-change document:

Manuscript page 5, 3rd paragraph

Manuscript page 6, 3rd paragraph

Manuscript pages 23-24, section on Model evaluating association between plasma pTau217 and kidney function

Supplementary Figure 3

Reply to reviewer 3

D1. It is important to point out that plasma measures cannot identify the spatial distribution of phosphorylated-tau. The spatial distribution of tau is essential for distinguishing tauopathies that are not Alzheimer's Disease (AD), including Frontotemporal Lobar Degeneration (FTLD), Chronic Traumatic Encephelopathy (CTE), and Primary Age-Related Tauopathy (PART), from AD. Data for the prevalence of these other tauopathies is even sparser than for AD, but these neurodegenerative patterns that are distinct from AD should be mentioned.

One early publication describing PART:

Crary, John F.; Trojanowski, John Q.; Schneider, Julie A.; Abisambra, Jose F.; Abner, Erin L.; Alafuzoff, Irina; Arnold, Steven E.; Attems, Johannes; Beach, Thomas G. (2014-12-01). "Primary age-related tauopathy (PART): a common pathology associated with human aging". *Acta Neuropathologica*. 128 (6): 755–766. doi:10.1007/s00401-014-1349-0. ISSN 0001-6322. PMC 4257842. PMID 25348064.

We agree with the reviewer that the regional information of molecular PET is an important indicator of observed or expected clinical symptoms.

Plasma - and also CSF - pTau217 reflects phosphorylated, soluble tau in the context of amyloid β pathology (Jack, *Alz Dementia* 2024, PMID 38934362), as well as serving as a marker of aggregated tau pathology. Elevated concentrations can occur decades before the onset of aggregated tau (Ossenkoppelse, *Lancet Neurology* 2022, PMID: 35643092). Plasma pTau217 has shown good discriminative accuracy for distinguishing between pathology-confirmed AD and FrontoTemporal Lobar Degeneration (FTLD) (Thijssen, *Lancet Neurology* 2021, PMID 34418401) and for confirming AD neuropathology in FTLD-related syndromes (VandeVrede, *JAMA Neurology* 2025, PMID 39928343). Plasma pTau217 has also showed fair ability to differentiate A β (+) Traumatic encephalopathy syndrome from A β (-) Traumatic encephalopathy syndrome (AUC = 0.79 [0.54-1.00], p = 0.04) (Asken, *Neurology* 2022, PMID: 35577574) and to differentiate AD from Primary Age-Related Tauopathy (AUC of 0.82) (Yu, *Acta Neuropathol.* 2023, PMID: 37031430).

Place of change in the track-change document:

Manuscript page 8, first and last paragraph, added text and more references, including the one suggested by the reviewer.

D2. The scientific rigor of this work in the study of biomarkers for AD would be substantially improved if the terminology were more precise throughout. The plasma-tau biomarkers do indeed show promising associations (AUC around 0.8) with measures from CSF or from neuropathology, but these findings from different biospecimens are not interchangeable. To their credit, the authors avoid the confusion between Alzheimer's Disease (AD) (defined as a biologic process) and AD-Dementia (the clinical recognition of dementia of the Alzheimer's type). However, describing plasma biomarkers as ADNC (Alzheimer's Disease Neuropathologic Change) is a step toward imprecision.

The agreement between antemortem high plasma pTau217 and the presence of significant amounts of plaques and tangles at post-mortem examination is very high. Palmqvist et al. (Palmqvist, *JAMA* 2020, PMID 32722745) found an AUC of 0.89 (95% CI, 0.81-0.97) and in secondary analyses that compared participants with high likelihood of AD vs non-AD, the AUC for antemortem plasma P-tau217 levels was 0.98 (95% CI, 0.94 to 1.00; 94% correctly classified). It is also worth pointing out that plasma pTau217 biomarkers are now approved to determine the presence of plaques in the brain in symptomatic people ages >55 (Schindler, *Nat Rev Neurolog.* 2024, PMID: 38866966). Plasma pTau217 is also listed in the research criteria as Core 1 A biomarker (Jack, *Alz Dementia*, 2024, PMID 38934362).

Still, we understand the reviewer's concern and acknowledge the importance of clearly defining the terminology employed. Thus, we have added the sentence 'For terminological clarity, in the remainder of this article, the term 'ADNC' refers specifically to the presence of elevated plasma pTau217 concentration, used as a surrogate marker for Alzheimer's disease neuropathologic changes.'

Place of change in the track-change document:

Abstract, line 4

Manuscript page 4, first paragraph

Specified the context of the term ADNC in this work also in the notes in Figure 1-3, Supplementary Figure 2 and 4 and Tables.

E1: I understand this study to be a nested cross-sectional study of participants in one of two waves (HUNT3, HUNT4) of a large Norwegian cohort study in which exactly one blood draw was analysed. Since 2,391 individuals participated in both HUNT3 and HUNT4; please clarify whether one blood biospecimen or two biospecimens were studied. I assume one biospecimen per participant, as I do not see any methodologic allowance for repeated ADp-TAU values

We thank the reviewer for pointing out that this was unclear. Most participants included in the HUNT3 analysis of this study provided two blood biospecimen, one at HUNT3 and one at HUNT4. The reason for us not analyzing blood biospecimen from all HUNT3 participants was due to funding limitations. To avoid potential selection bias in the results, we used an inverse probability weighting approach for the HUNT3 sample when doing the statistical analysis.

Place of change in the track-change document:

Manuscript page 20, section on Cohort Selection and Study Design

Manuscript page 22, section on Selection Bias and Weighting

E2: Tight control for age (at least in the 5-year intervals, but ideally in 1-year intervals) is essential when comparing sex or diagnosis groups or medical comorbidities.

We thank the reviewer for this important consideration. We have re-analyzed the association between self-reported medical disorders and plasma pTau217 above the upper cut-off (≥ 0.63 pg/mL) with adjustment for age (both 5-year intervals and 1-year intervals), sex, APOE $\epsilon 4$ allele count, cognition, serum creatinine, and education level. Applying this, there was no longer a significant association between any comorbidity and pTau217 ≥ 0.63 pg/mL, neither when comparing ≥ 0.63 pg/mL to < 0.63 pg/mL, nor when comparing ≥ 0.63 pg/mL to < 0.40 pg/mL groups.

Place of change in the track-change document:

Manuscript page 7, 3rd paragraph

Manuscript page 9, 2nd paragraph

Manuscript page 20, last paragraph

Supplementary Table 7

E3: Figure 2: Age is clear confounding variable in the comparison of the diagnosis groups, as the diagnosis distributions shift with age. Consider repeating Figure 1 separately for each of the 3 diagnosis groups; such figures might well improve the clarity of the comparison. would be an improvement.

We have now modified Figure 2 to show the 3 diagnosis groups in each 5-year age group, as requested.

Place of change in the track-change document:

Manuscript page 17, Figure 2

E4. As described above (point D), I recommend removing “ADNC” from most of the manuscript, with the possible exception of discussion.

Please refer to response D2, which we trust addresses the reviewer’s concern.

Place of change in the track-change document:

Abstract, line 4

Manuscript page 4, first paragraph

Specified the context of the term ADNC in this work also in the notes in Figure 1-3, Supplementary Figure 2 and 4 and Tables.

E5. Minor point: A brief explanation in the supplementary material of the reason a very small number of persons were not classified into a cognitive status. Does this refer to the inability to get a consensus diagnosis? Or patterns of dementia not consistent with AD? Or incomplete testing?

We thank the reviewer for this relevant question. Of the 8,949 HUNT4 70+ participants with available blood samples, 153 could not be classified as cognitively unimpaired, MCI, or dementia. This was due to missing information on cognitive status – caused by either refusal to undergo cognitive testing or lack of available caregiver information. We have added this information in the Supplementary Material.

Place of change in the track-change document:

New Supplementary Material_Assessment of cognition, physical performance, anxiety, depression, neuropsychiatric symptoms and activities of daily living

G1. See earlier points on the distinction between biomarker positivity and ADNC

Please refer to response D2, which we trust addresses the reviewer’s concern.

Place of change in the track-change document:

Abstract, line 4

Manuscript page 4, first paragraph

Specified the context of the term ADNC in this work also in the notes in Figure 1-3, Supplementary Figure 2 and 4 and Tables.

G2. The vascular risk factors could be discussed in the context of vascular disease, especially with the recognition that vascular disease can both cause dementia and increase the odds that AD pathology manifests as AD dementia.

We agree with the reviewer that it is important to recognize that vascular disease can both cause vascular dementia and increase the odds that AD pathology manifests as AD dementia. After re-analyzing the association between self-reported medical disorders and plasma pTau217 above the upper cut-off (≥ 0.63 pg/mL) with adjustment for age (both 5-year intervals and 1-year intervals), sex, APOE $\epsilon 4$ allele count, cognition, serum creatinine, and education level there was no longer a significant association with vascular disease. We have not examined the possible synergistic effect of ADNC and comorbidities on cognition in our material, which would be interesting to examine in a separate paper. We have added a comment in the discussion about the relevance of vascular factors.

Place of change in the track-change document: Manuscript page 7, 3rd paragraph

G3. The authors recognize the potential differences of different ethnic and racial groups, and recognize that these non-Caucasian groups are not well-represented in Norway. One subtle change would be to refer to “populations” rather than “population” throughout the introduction and conclusion. A rigorous evaluation of one population is very valuable. The Norwegian population may differ even from other Caucasian populations in many ways, including in the presence of risk factors for vascular disease. Since mixed pathologies are important in the manifestation of dementia in older persons, this clear picture of Norway may not automatically translate to other countries.

We thank the reviewer for an important consideration. We have exchanged "population" with "populations" where appropriate and have added "Norwegian" where indicated.

Place of change in the track-change document:

Abstract, line 4

Manuscript page 3, first paragraph

Manuscript page 6, 3rd paragraph

Manuscript page 9, 2nd paragraph and last paragraph